# TPL-2 kinase induces phagosome acidification to promote macrophage killing of bacteria

Felix Breyer[1] iD, Anetta Härtlova[2,3] iD, Teresa Thurston[4] iD, Helen R Flynn[1] iD, Probir Chakravarty[1] iD, Julia Janzen[1], Julien Peltier[2], Tiaan Heunis[2], Ambrosius P Snijders[1], Matthias Trost[2] iD & Steven C Ley[1,5,*] iD

## Abstract

Tumour progression locus 2 (TPL-2) kinase mediates Toll-like receptor (TLR) activation of ERK1/2 and p38α MAP kinases in myeloid cells to modulate expression of key cytokines in innate immunity. This study identified a novel MAP kinase-independent regulatory function for TPL-2 in phagosome maturation, an essential process for killing of phagocytosed microbes. TPL-2 catalytic activity was demonstrated to induce phagosome acidification and proteolysis in primary mouse and human macrophages following uptake of latex beads. Quantitative proteomics revealed that blocking TPL-2 catalytic activity significantly altered the protein composition of phagosomes, particularly reducing the abundance of V-ATPase proton pump subunits. Furthermore, TPL-2 stimulated the phosphorylation of DMXL1, a regulator of V-ATPases, to induce V-ATPase assembly and phagosome acidification. Consistent with these results, TPL-2 catalytic activity was required for phagosome acidification and the efficient killing of *Staphylococcus aureus* and *Citrobacter rodentium* following phagocytic uptake by macrophages. TPL-2 therefore controls innate immune responses of macrophages to bacteria via V-ATPase induction of phagosome maturation.

**Keywords** DMXL1; macrophage phagosome; *Staphylococcus aureus*; TPL-2 kinase; V-ATPase

**Subject Categories** Immunology; Signal Transduction

The EMBO Journal (2021) 40: e106188

## Introduction

A key component of the innate immune response involves the killing of phagocytosed microbes, such as bacteria, by macrophages (Pauwels *et al*, 2017). Phagocytosis internalises bacteria into membrane-bound vacuoles in the cytoplasm called phagosomes (Flannagan *et al*, 2012). The nascent phagosome is innocuous, and to kill internalised bacteria, the phagosome must mature via an ordered series of membrane fusion and fission events with endosomes and ultimately with lysosomes to form the phagolysosome, a potent microbicidal organelle (Flannagan *et al*, 2009). During the maturation process, phagosomes become increasingly acidic, highly oxidative and enriched with proteases and hydrolases that can degrade the internalised bacteria (Flannagan *et al*, 2009). Phagosome acidification is mediated by V-ATPases, ATP-dependent proton pumps, which are recruited to the phagosome during maturation (Lukacs *et al*, 1990; Sun-Wada *et al*, 2009). Acidification inhibits bacterial growth (Downey *et al*, 1999; Ip *et al*, 2010), activates cathepsin proteases that degrade internalised bacteria (Yates *et al*, 2005) and promotes NOX2 production of reactive oxygen species (ROS) that damage bacterial proteins, lipids and nucleic acids (Savina *et al*, 2006).

Bacterial infection also engages macrophage Toll-like receptors (TLRs) on the plasma and endosomal membranes, triggering intracellular signalling pathways that control activation of NF-κB transcription factors and each of the major mitogen-activated protein (MAP) kinases (extracellular signal-regulated kinases 1 and 2 [ERK1/2], Jun amino-terminal kinases 1 and 2 [JNK1/2] and p38α) (Arthur & Ley, 2013). Together, these stimulate the expression of multiple genes, the products of which can directly target the invading bacterium (e.g. antimicrobial peptides and NOS2) or induce recruitment of additional immune cells (e.g. cytokines and chemokines) (Smale, 2010; Kawasaki & Kawai, 2014). This inflammatory response is essential for killing of bacteria by macrophages and also the subsequent induction of the adaptive immune response (Nau *et al*, 2002).

TLR activation of ERK1/2 and p38α in macrophages is mediated by TPL-2 (tumour progression locus 2, also known as MAP3K8), a MAP 3-kinase that directly phosphorylates and activates the MAP 2-kinases MKK1, 2, 3 and 6 (Gantke *et al*, 2012; Pattison *et al*, 2016). TPL-2 is critical for inflammatory immune responses to invading

1 The Francis Crick Institute, London, UK
2 Biosciences Institute, Newcastle University, Newcastle-upon-Tyne, UK
3 Wallenberg Centre for Molecular and Translational Medicine, University of Gothenburg, Gothenburg, Sweden
4 Department of Infectious Diseases, MRC Centre for Molecular Bacteriology & Infection, Imperial College London, London, UK
5 Department of Immunology & Inflammation, Centre for Molecular Immunology & Inflammation, Imperial College London, London, UK
*Corresponding author. Tel: +44 203 796 2207; E-mail: s.ley@imperial.ac.uk

bacteria, fungi and viruses. TPL-2 forms a stoichiometric complex with NF-κB1 p105 and A20 binding inhibitor of NF-κB (ABIN)-2 in unstimulated macrophages (Lang *et al*, 2004). NF-κB1 p105 functions as an inhibitor of TPL-2 and TLR activation of ERK1/2, and p38α requires the liberation of TPL-2 from p105-mediated inhibition (Beinke *et al*, 2003, 2004). This results from p105 proteolysis by the proteasome triggered by IκB kinase (IKK)-induced p105 phosphorylation (Belich *et al*, 1999). ABIN-2 is also released from p105 after TLR stimulation but is not required for TPL-2 activation, and its function in innate immune responses remains unclear.

TPL-2 expression is required for efficient immune responses to *Citrobacter rodentium*, decreasing the bacterial burden and dissemination to the liver and spleen (Acuff *et al*, 2017). TPL-2 deficiency reduces neutrophil recruitment to the colon at peak infection. *In vitro* experiments have shown that TPL-2 expression in neutrophils is required for LPS activation of ERK1/2 and induction of tumour necrosis factor (TNF) secretion, similar to that in macrophages (Acuff *et al*, 2017). TPL-2 is also required for efficient killing of phagocytosed *C. rodentium*, although phagocytic uptake of bacteria is TPL-2-independent (Acuff *et al*, 2017). However, the mechanism by which TPL-2 regulates bacterial killing has not been established. A recent study by one of our groups using quantitative mass spectrometry demonstrated that TPL-2 catalytic activity potentially regulates intracellular trafficking. $Tpl2^{D270A}$ kinase-dead mutation reduced the phosphorylation of several proteins linked to endocytosis, vesicle trafficking and GTPase signalling in LPS-stimulated macrophages (Pattison *et al*, 2016). Together, these results raised the interesting possibility that TPL-2 promotes the killing of internalised bacteria by controlling phagosome maturation in myeloid cells.

In the present study, the potential regulation of phagosome maturation by TPL-2 in macrophages was investigated. TPL-2 catalytic activity was found to induce macrophage phagosome proteolytic activity and phagosome acidification, both independently of its ability to activate MAP kinase signalling. These effects were mediated by direct regulation of V-ATPase assembly and function via controlling the phosphorylation of the V-ATPase-associated regulatory protein DMXL1. Consistent with these findings, TPL-2 catalytic activity was necessary for efficient macrophage killing of phagocytosed *Staphylococcus aureus* and *Citrobacter rodentium*. These results indicate that TPL-2 has two key functions in the innate immune response of macrophages to bacterial infection. TPL-2 mediates TLR activation of MAP kinase signalling to control inflammation by modulation of gene expression and directly promotes bacterial killing by MAP kinase-independent induction of phagosome acidification.

# Results

### TPL-2 catalytic activity promotes macrophage phagosome maturation

To investigate the role of TPL-2 signalling in phagosome maturation, bone marrow-derived macrophages (BMDMs) were generated from $Tpl2^{D270A/D270A}$ knock-in mice, which express catalytically inactive TPL-2 (Sriskantharajah *et al*, 2014). Consistent with earlier experiments using $Tpl2^{-/-}$ neutrophils (Acuff *et al*, 2017), phagocytic uptake of fluorescently labelled beads by $Tpl2^{D270A/D270A}$ BMDMs

was similar to wild-type (WT) controls (Fig 1A). To investigate whether TPL-2 regulated phagosome maturation, bulk intra-phagosomal proteolysis was monitored using latex beads coupled to the fluorescent substrate DQ green BSA (Yates & Russell, 2008; Russell *et al*, 2009). Alexa Fluor 594 labelling of beads was used to normalise phagocytic uptake. Real-time measurements showed that phagosome proteolysis was significantly reduced in $Tpl2^{D270A/D270A}$ BMDMs compared to WT controls (Fig 1B).

Next, a confocal microscopy assay was used to monitor cathepsin protease activation following phagocytic uptake of latex beads by $Tpl2^{D270A/D270A}$ BMDMs compared to WT control cells, using a fluorescently labelled cathepsin L target peptide. Thirty mins after bead uptake, phagosome cathepsin activity was significantly reduced by $Tpl2^{D270A}$ mutation (Fig 1C). These results showed that TPL-2 catalytic activity is required to increase protease activity inside phagosomes of primary macrophages.

During maturation, the lumens of phagosomes become increasingly acidic due to the action of V-ATPases that are recruited by fusion of phagosomes initially with endocytic vesicles and ultimately lysosomes (Kinchen & Ravichandran, 2008). Intra-phagosomal acidification was assayed using latex beads coupled to the pH indicator BCECF (Russell *et al*, 1995). Real-time measurements showed that phagosomal acidification was significantly decreased in $Tpl2^{D270A/D270A}$ BMDMs compared to WT BMDMs (Fig 1D). $Tpl2^{D270A}$ mutation reduced phagosome acidification to a similar degree to the pre-treatment of WT BMDMs with the V-ATPase inhibitor bafilomycin A1 (Fig 1D). Consistent with these results, analysis of phagosomal pH by staining with LysoTracker Red, a dye that labels acidic compartments, and confocal microscopy demonstrated that $Tpl2^{D270A}$ mutation reduced LysoTracker Red staining following latex bead uptake (Fig 1E).

The production of reactive oxygen species (ROS) in phagosomes by NAPDH oxidase is essential for killing internalised bacteria and is dependent on an intra-phagosomal proton gradient by V-ATPases (Mantegazza *et al*, 2008). Consistent with its inhibitory effects on phagosome acidification, $Tpl2^{D270A}$ mutation significantly reduced ROS generation in BMDMs after uptake of latex beads coupled to the fluorescence reporter substrate OxyBURST Green BSA (Fig 1F) (Vanderven *et al*, 2009). $Tpl2^{D270A}$ mutation also significantly reduced ROS generation detected microscopically using ROS Deep Red dye following latex bead uptake (Fig 1G).

Together, these results show that TPL-2 catalytic activity promotes the maturation of phagosomes in macrophages.

### TPL-2 induces phagosome maturation independently of MAP kinase activation

TPL-2's established functions in innate immunity are mediated by activation of MAP kinase pathways (Gantke *et al*, 2011). However, uptake of latex beads by BMDMs did not induce ERK1/2 and p38α activation, as detected by phospho-antibody immunoblotting, suggesting that TPL-2 modulated phagosome maturation independently of its ability to activate MAP kinases. Genetic and pharmacological experiments were used to investigate whether MAP kinase activation was required for TPL-2 catalytic activity to promote phagosome maturation.

IKK triggers proteolysis of NF-κB1 p105 by phosphorylating Ser930 and Ser935 in the p105 PEST region (Lang *et al*, 2003).

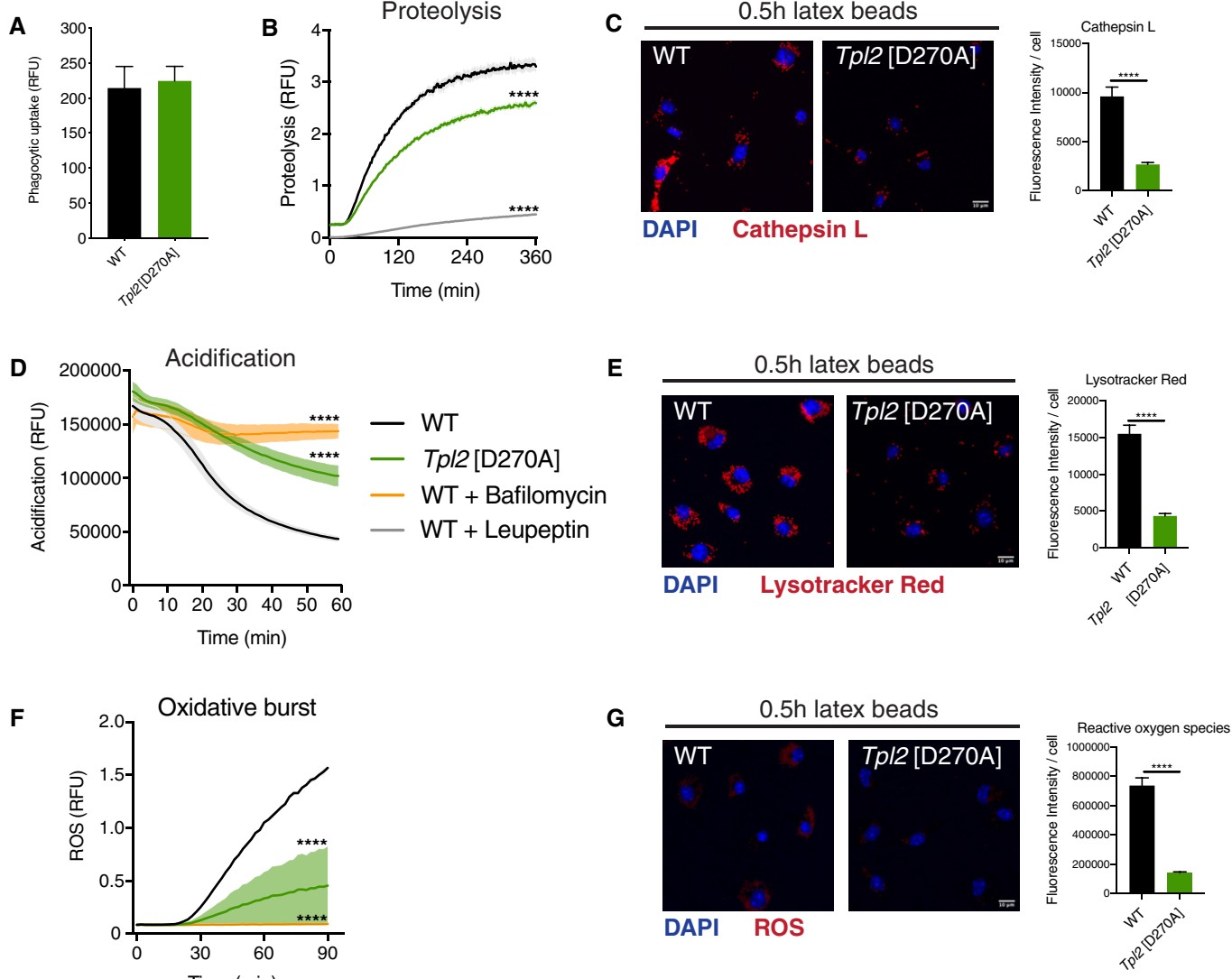

**Figure 1. *Tpl2*[D270A] mutation impairs phagosome maturation.**

A  Phagocytic uptake of fluorescently labelled latex beads by WT and *Tpl2*[D270A] (*Tpl2*[D270/D270A]) BMDMs. Intracellular fluorescence was monitored following uptake of AF488 latex beads by BMDMs (*n* = 4 wells).

B  Intra-phagosomal proteolysis in WT and *Tpl2*[D270A] BMDMs was assayed following uptake of DQ Green BSA / AF594 latex beads. As positive assay control, BMDMs were separately pre-treated with 100 μg/ml leupeptin for 1 h to inhibit serine-cysteine proteases (*n* = 4 wells).

C  Cathepsin activity assay in WT and *Tpl2*[D270A] BMDMs 0.5 h after uptake of latex beads. BMDMs were stained with the Magic Red cathepsin L substrate (red). Average fluorescence intensity of the cathepsin probe per cell was quantified (*n* = 40–51 cells).

D  Intra-phagosomal acidification in WT and *Tpl2*[D270A] BMDMs was monitored following uptake of BCECF-coupled latex beads. BMDMs were pre-treated with 1 μM bafilomycin A1 for 15 min to inhibit V-ATPases (*n* = 4 wells).

E  pH assay in WT and *Tpl2*[D270A] BMDMs upon 0.5 h after incubation with latex beads. BMDMs were stained with the LysoTracker Red DND-99 dye (red). Average fluorescence intensity of the LysoTracker Red probe per cell was quantified (*n* = 95–126 cells).

F  Intra-phagosomal oxidative burst in WT and *Tpl2*[D270A] BMDMs was assayed following uptake of OxyBURST Green BSA / AF594 latex beads. BMDMs were pre-treated with 1 μM bafilomycin A1 for 15 min to inhibit ROS production (*n* = 4 wells).

G  Reactive oxygen species assay in WT and *Tpl2*[D270A] BMDMs upon 0.5 h after incubation with latex beads. BMDMs were stained with the ROS Deep Red dye. Average fluorescence intensity of the ROS Deep Red probe per cell was quantified (*n* = 80–110 cells).

Data information: One representative experiment out of three shown. Error bars and shaded areas represent SEM. ****$P < 0.0001$. Panels (B, D, F) Paired Mann–Whitney *t*-test; All differences relative to WT are ****. Panels (C, E, G) Student's unpaired *t*-test.

$Nfkb1^{SSAA}$ mutation, which changes both serines to alanine, blocks IKK-induced p105 proteolysis, thereby preventing release of TPL-2 and ABIN-2 from p105 and blocking TPL-2 activation of ERK1/2 and p38α MAP 2 kinases (Yang *et al*, 2012; Pattison *et al*, 2016). Phagosome proteolysis and acidification, however, were not affected by $Nfkb1^{SSAA}$ mutation (Fig 2A and B). Importantly, however, both

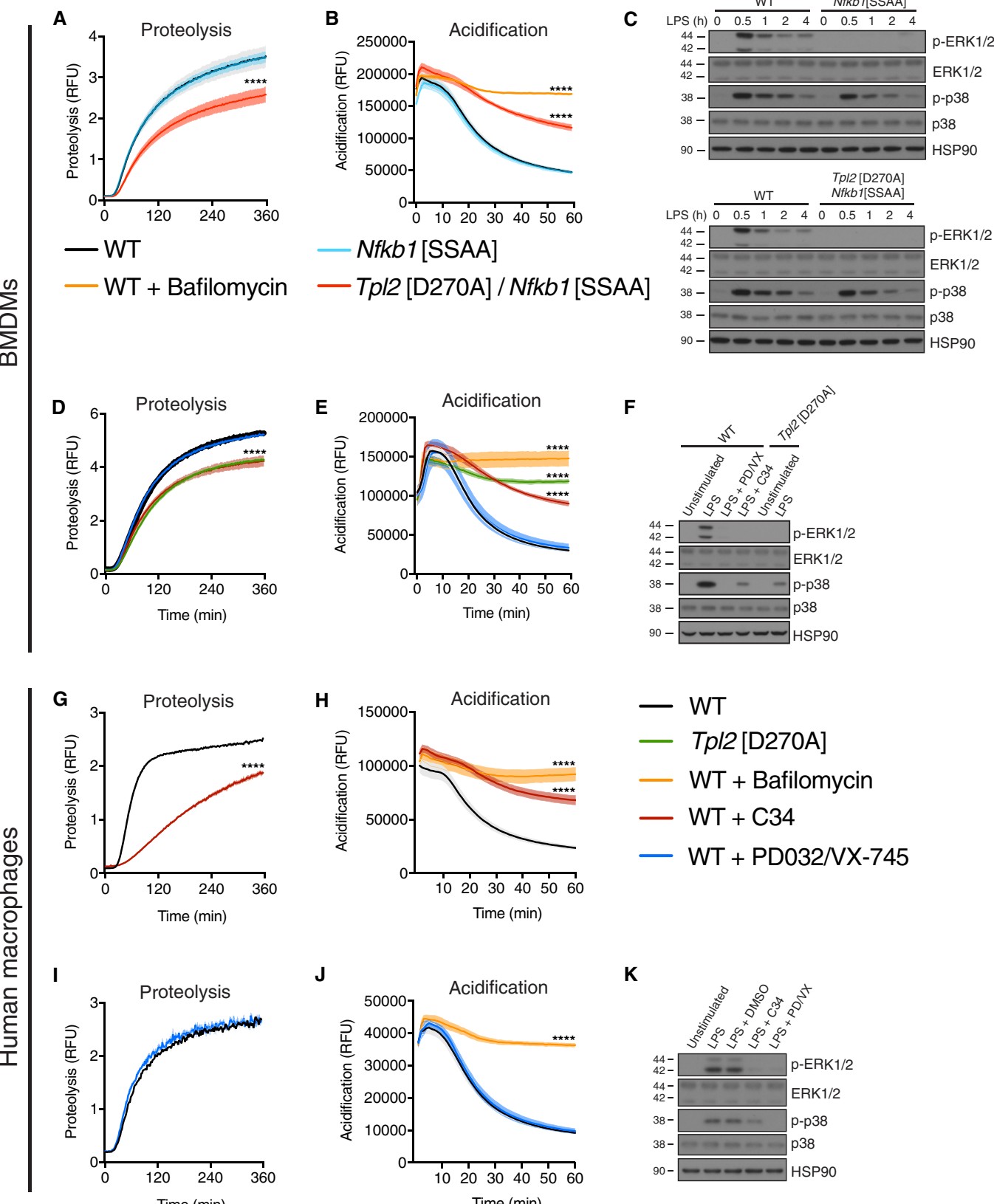

**Figure 2.**

**Figure 2. *Tpl2*[D270A] mutation promotes phagosome maturation independently of MAP kinase activation.**

A–F   Experiments were performed using murine BMDMs. (A) Intra-phagosomal proteolysis in WT, *Nfkb1*[SSAA] (*Nfkb1*[SSAA/SSAA]), and *Nfkb1*[SSAA]/*Tpl2*[D270A]
      BMDMs were monitored as in Fig 1B (*n* = 4 wells). (B) Intra-phagosomal acidification in WT, *Nfkb1*[SSAA], and *Nfkb1*[SSAA]/*Tpl2*[D270A] BMDMs was
      monitored as in Fig 1C (*n* = 4 wells). (C) Cell extracts from LPS-stimulated *Nfkb1*[SSAA] and *Nfkb1*[SSAA]/*Tpl2*[D270A] BMDMs were immunoblotted for the
      indicated antigens. (D) Intra-phagosomal proteolysis in WT and inhibitor-treated BMDMs (*n* = 4 wells) was monitored as in Fig 1B. BMDMs were pre-
      treated with 0.1 μM PD0325901 (10 min) to inhibit MEK1, pre-treated with 1 μM VX-745 (1 h) to inhibit p38α, and pre-treated with 10 μM C34 (1 h) to
      inhibit TPL-2. (E). Intra-phagosomal acidification in WT and inhibitor-treated BMDMs (see (D) for conditions) was assayed as in Fig 1C (*n* = 4 wells). (F)
      Cell extracts from LPS-stimulated, inhibitor-treated WT, and *Tpl2*[D270A] BMDMs were immunoblotted for the indicated antigens. 1 μM bafilomycin A1 was
      added to inhibit V-ATPases.
G–K   Experiments were performed using human primary monocyte-derived macrophages. (G) Intra-phagosomal proteolysis in human macrophages was monitored
      following uptake of DQ Green BSA / AF594 latex beads. TPL-2 catalytic kinase activity was blocked by pre-treatment with 10 μM C34 for 1 h (*n* = 4 wells). (H)
      Intra-phagosomal acidification in human macrophages was monitored following uptake of BCECF-coupled latex beads (*n* = 4 wells). (I) Intra-phagosomal
      proteolysis in human macrophages was assayed as in Fig 2G. MAP kinase activity was inhibited by combinatorial pre-treatment with 0.1 μM PD0325901 (MEK1
      inhibition) for 10 min and 1 μM VX-745 (p38 inhibition) for 1 h (*n* = 4 wells). (J) Intra-phagosomal acidification in human macrophages was monitored as in
      Fig 2H (*n* = 4 wells). (K) Cell extracts from LPS-stimulated, inhibitor-treated primary human macrophages were immunoblotted for p-ERK1/2, ERK1/2, p-p38, p38,
      and HSP90.

Data information: One representative experiment out of three shown. Error bars and shaded areas represent SEM. ****$P < 0.0001$. Panels (A, B, D, E, G, H, I, J) Paired
Mann–Whitney *t*-test; all differences relative to WT are ****.

processes were reduced by *Nfkb1*[SSAA] *Tpl2*[D270A] compound mutation confirming their dependence on TPL-2 catalytic activity (Fig 2A and B). Inhibition of ERK1/2 phosphorylation in *Nfkb1*[SSAA] and *Nfkb1*[SSAA] *Tpl2*[D270A] BMDMs following LPS stimulation was confirmed by immunoblotting (Fig 2C). In line with these genetic data, simultaneous pharmacological inhibition of ERK1/2 (PD0325901) and p38α (VX-745) activity did not alter phagosome proteolysis or acidification (Fig 2D and E). In contrast, TPL-2 inhibition with the small molecule inhibitor C34 (Wu *et al*, 2009) significantly reduced both processes to a similar degree as *Tpl2*[D270A] mutation (Fig 2D and E). Substantial reductions of MAP kinase activation by pharmacological inhibitors were confirmed by immunoblotting of LPS-stimulated BMDM lysates (Fig 2F).

TPL-2 catalytic activity therefore stimulated phagosome maturation in macrophages independently of MAP kinase activation (via unknown substrates). The inhibitory effects of C34 on phagosome proteolysis and acidification in WT BMDMs also demonstrated that the effects of TPL-2 catalytic activity on phagosome maturation were mediated acutely following latex bead uptake.

To investigate the role of TPL-2 in phagosome maturation in human primary macrophages, macrophages were generated from monocytes isolated from peripheral blood by culturing in GM-CSF and phagosome maturation was assayed as for BMDMs. C34 inhibition of TPL-2 catalytic activity reduced intra-phagosomal proteolysis (Fig 2G) and acidification (Fig 2H) in monocyte-derived human macrophages compared to cells pre-treated with vehicle control. Pharmacological inhibition of MEK1/2 (PD0325901) (Ciuffreda *et al*, 2009) and p38α (VX-745) (Duffy *et al*, 2011) did not alter either phagosome proteolysis (Fig 2I) or acidification (Fig 2J), although these inhibitors potently inhibited phosphorylation of ERK1/2 and p38α following LPS stimulation (Fig 2K). These experiments showed that MAP kinase-independent stimulation of phagosome maturation by TPL-2 catalytic activity was conserved between mouse and human primary macrophages.

## TPL-2 catalytic activity regulates the protein composition of phagosomes

To further study the molecular mechanisms by which TPL-2 kinase activity promotes phagosome maturation, the composition of

phagosomes isolated from *Tpl2*[D270A/D270A] and WT BMDMs following uptake of latex beads was analysed by mass spectrometry. *Tpl2*[D270A] mutation significantly altered the composition of the phagosome proteome (Fig 3A and B, Dataset EV1). Phagosome abundance of numerous Rab GTPases was decreased by *Tpl2*[D270A] mutation (Fig 3A). These included RAB5 and RAB7, which have key roles in regulating the maturation of phagosomes (Vieira *et al*, 2003; Huynh *et al*, 2007). The abundance of LAMP1, which is essential for late phagosome/lysosome fusion, was also decreased in *Tpl2*[D270A/D270A] phagosomes compared with WT cells (Fig 3A and D). The levels of seven V-ATPase subunits, including V-ATPase subunits D, D1 and E1, were also significantly reduced (Fig 3C). This finding was particularly interesting given the inhibitory effect of *Tpl2*[D270A] mutation on phagosome acidification. Phagosome abundance of four members of the LAMTOR protein complex, a key regulator of lysosomal trafficking (Colaço & Jäättelä, 2017), was also reduced in *Tpl2*[D270A/D270A] BMDMs compared to controls. Immunoblotting confirmed that *Tpl2*[D270A] mutation reduced levels of RAB5 GTPase and LAMP-1 associated with purified phagosomes (Fig 3D). The effect of TPL-2 catalytic activity on the abundance of phagosomal proteins was selective since abundance of numerous proteins on phagosomes, including vimentin, SNARE, vacuolar protein sorting (VPS) and ESCRT proteins, was similar between WT and *Tpl2*[D270A/D270A] BMDMs (Figs 3D and EV1).

Consistent with these changes in phagosome composition, gene set enrichment analyses to identify significantly downregulated biological processes revealed that *Tpl2*[D270A] mutation decreased the abundance of proteins involved in vesicle-mediated transport, receptor-mediated transport, ion transport, vesicle proteolysis and regulation of pH (Fig 3B). These processes are all important in the generation of mature phagolysosomes (Flannagan *et al*, 2009; Pauwels *et al*, 2017).

Together, these experiments indicated that blocking TPL-2 catalytic activity substantially changed the composition of phagosomes in BMDMs, reducing the abundance of several proteins with key roles in regulating different stages of phagosome maturation. Importantly, this analysis revealed that TPL-2 catalytic activity promoted the recruitment of V-ATPase subunits to phagosomes during their maturation suggesting a possible mechanism for TPL-2 control of phagosome acidification.

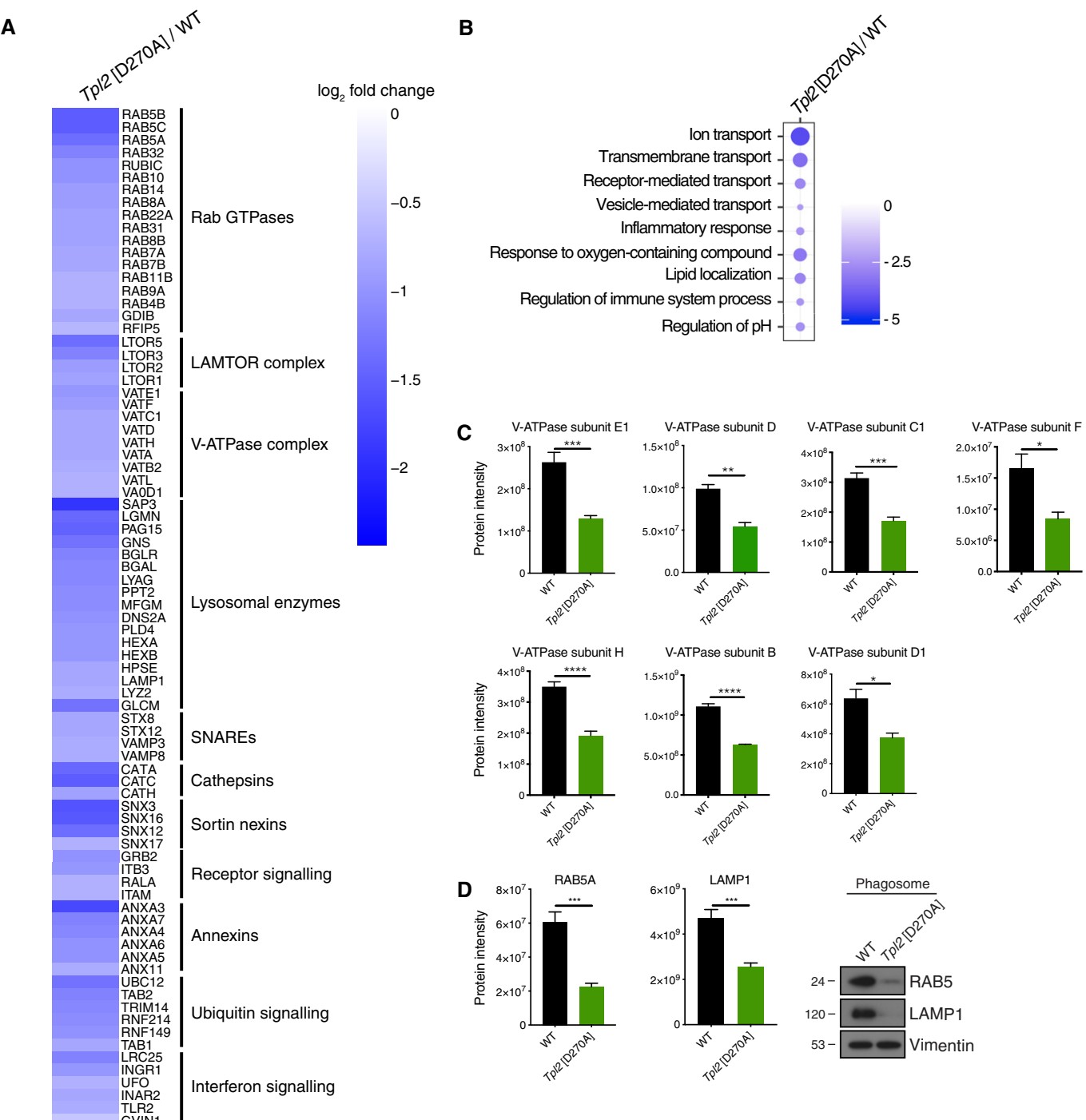

**Figure 3.** *Tpl2*[D270A] mutation alters the protein composition of phagosomes.

WT and *Tpl2*[D270A] BMDMs were incubated with latex beads for 0.5 h. Latex bead phagosomes were purified from *Tpl2*[D270A] and WT BMDMs and analysed by mass spectrometry. Biological triplicates were analysed for each genotype.

A  Heatmap of selected proteins that were significantly downregulated in BMDMs from *Tpl2*[D270A] mice relative to WT ($P < 0.05$). Selected hits were grouped into clusters according to their molecular functions.

B  Gene set enrichment analysis (GSEA) of significantly downregulated biological processes in phagosomal fractions. Changes of *Tpl2*[D270A] phagosomes relative to WT. Dot colour; enrichment score. Dot size; statistical significance.

C  Protein intensities of V-ATPase subunits from phagosomes purified from WT and *Tpl2*[D270A] BMDMs, ($n = 3$ biological replicates).

D  Protein intensities of RAB5 and LAMP-1 from phagosome proteome analysis of WT and *Tpl2*[D270A] BMDMs ($n = 3$ biological replicates) (left). Immunoblot of isolated phagosomes from WT and *Tpl2*[D270A] BMDMs probed for RAB5, LAMP-1, and vimentin. Phagosomal fractions of two biological replicates were pooled. One representative experiment out of two shown (right) ($n = 2$).

Data information: Data were analysed by Student's *t*-test. Error bars represent SEM. *$P < 0.05$, **$P < 0.01$, ***$P < 0.001$, ****$P < 0.0001$.

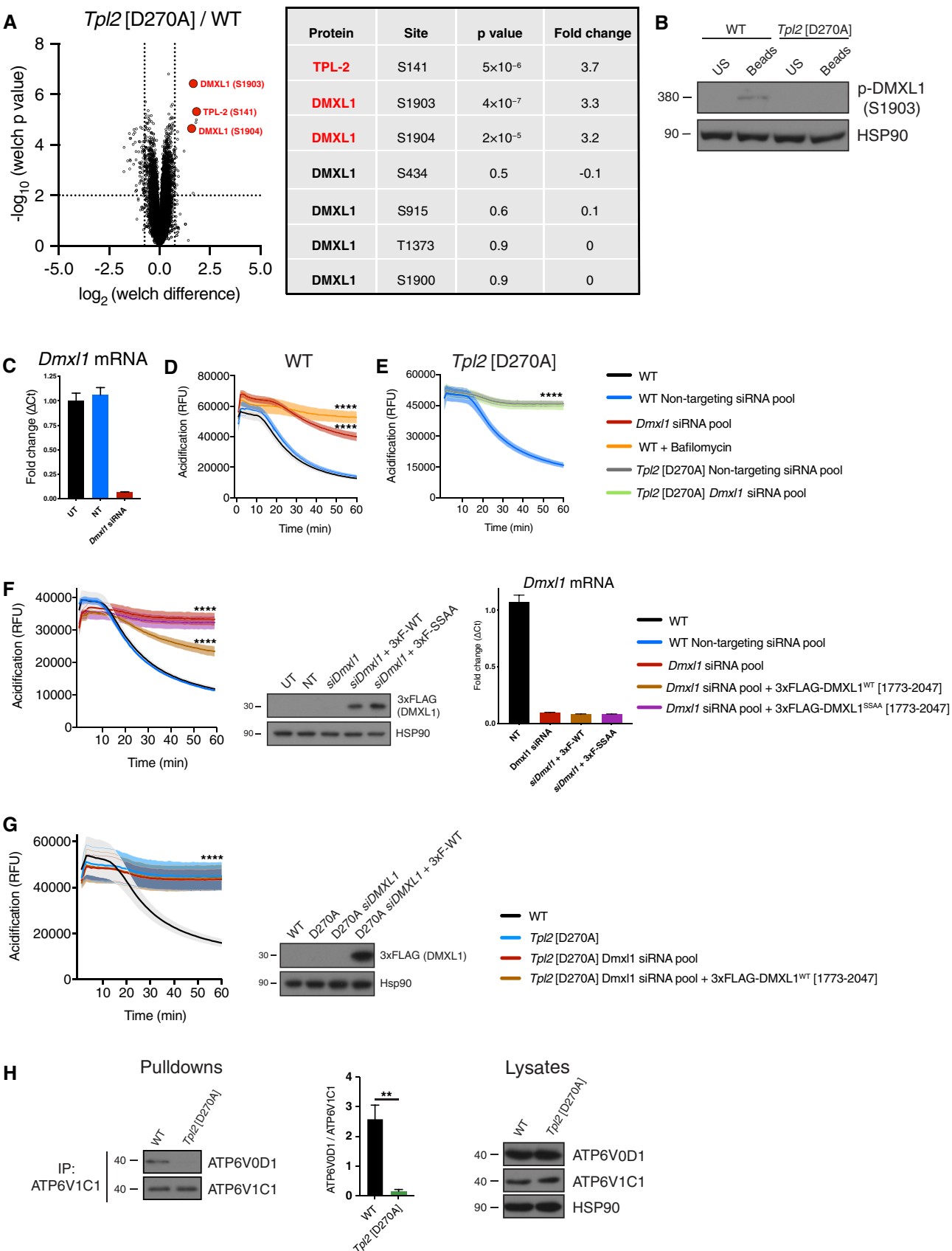

Figure 4.

**Figure 4. TPL-2 induces phosphorylation of DMXL1, a V-ATPase regulatory protein.**

A TPL-2-dependent phosphoproteome following phagocytosis of latex beads (0.5 h) was determined by TMT mass spectrometry. Volcano plot representing the significance (-log$_{10}$ $P$-values after Welch's $t$-test) versus phosphorylation fold change (Welch difference ratios) between WT and $Tpl2$[D270A] BMDMs. Five biological replicates were analysed per genotype ($n = 5$). Three of the most highly and significantly downregulated phospho-sites in $Tpl2$[D270A] BMDMs relative to WT, as well as unaltered DMXL1 phospho-sites, are shown.

B Total cell lysates from WT and $Tpl2$[D270A] BMDMs 0.5 h after incubation with latex beads were immunoblotted for phospho-DMXL1 (S1903) and HSP90 (loading control).

C–F $Dmxl1$ was knocked down in WT iBMDMs or $Tpl2$[D270A] iBMDMs by RNA interference using a SMARTpool ON-TARGETplus siRNA for 48 h. ON-TARGETplus non-targeting pool functioned as siRNA control. (C) qRT–PCR analysis of RNA extracted from iBMDMs was used to check the efficiency of $Dmxl1$ knockdown (D, E). $Dmxl1$ mRNA levels were normalised to $Hprt$ mRNA levels and fold changes calculated ($\Delta C_t$ values) ($n = 4$). (D) Intra-phagosomal acidification was assayed following uptake of BCECF-coupled latex beads by WT iBMDMs. As a control, BMDMs were pre-treated with 1 µM bafilomycin A1 for 15 min to directly block V-ATPase function ($n = 4$ wells). (E) Intra-phagosomal acidification was assayed following uptake of BCECF-coupled latex beads by $Tpl2$[D270A] iBMDMs ($n = 4$ wells). (F) Simultaneous with $Dmxl1$ siRNA knockdown, WT iBMDMs were co-transfected with plasmids expressing either 3xFLAG-DMXL1 (1,773–2,047) or 3xFLAG-DMXL1 S1903A/S1904A (1,773–2,047). Intra-phagosomal acidification was monitored as in Fig 4E ($n = 4$ wells). Immunoblot analysis of total cell lysates for FLAG demonstrated similar expression levels of the two DMXL1 polypeptides in iBMDMs. qRT–PCR analysis of RNA extracted from iBMDMs confirmed that $Dmxl1$ mRNA was efficiently knocked down ($n = 4$) (right) in cells expressing either 3xFLAG-DMXL1 (1,773–2,047) or 3xFLAG-DMXL1 S1903A/S1904A (1,773–2,047).

G Simultaneous with $Dmxl1$ siRNA knockdown, $Tpl2$[D270A] iBMDMs were co-transfected with a plasmid expressing 3xFLAG-DMXL1 (1,773–2,047). Intra-phagosomal acidification was monitored ($n = 4$ wells) (left). Immunoblot analysis of total cell lysates for FLAG demonstrated strong of 3xFLAG-DMXL1 (1,773–2,047) in iBMDMs (right).

H WT and $Tpl2$[D270A] BMDMs were treated with latex beads (1:50) for 30 min, and ATP6V1C1 was immunoprecipitated from cell extracts. Immunoblot analysis of eluates for ATP6V1C1 and co-immunoprecipitated ATP6V0D1 (left). ATP6V1C1 binding to ATP6V0D1 was quantified from three independent experiments (centre, $n = 3$). Immunoblot analysis of total cell lysates for both V-ATPase subunits and HSP90 (right).

Data Information: (B–H) One representative experiment out of three shown. Error bars and shaded areas represent SEM. ****$P < 0.0001$. Panels (D–G) Paired Mann–Whitney $t$-test. All differences relative to WT are ****. US, unstimulated; UT, untransfected; NT, non-targeting siRNA pool. Panel (H) Student's unpaired $t$-test. **$P < 0.01$.

## TPL-2 induces phosphorylation of DMXL1, a component of the V-ATPase complex

To further investigate the mechanism by which TPL-2 kinase activity regulated phagosome maturation, the phosphoproteome in $Tpl2$[D270A/D270A] and WT BMDMs was characterised 30 min after phagocytic uptake of latex beads. TMT (tandem mass tag) labelling and mass spectrometry were used to quantify differences in phosphopeptide abundance (Dataset EV2).

The phospho-site most highly downregulated by $Tpl2$[D270A] mutation was Ser141 on TPL-2 itself (Fig 4A), a known autophosphorylation site (Stafford *et al*, 2006; Xu *et al*, 2018). Two of the next most downregulated phospho-sites were Ser1903 and Ser1904 of DMXL1, which were both reduced over threefold. Phosphorylation of DMXL1 on Ser434, Ser915, Thr1373 and Ser1900 was not altered by $Tpl2$[D270A] mutation indicating that TPL-2 catalytic activity did not change DMXL1 protein abundance (Fig 4A).

A phospho-Ser1903 DMXL1-peptide antibody was generated and immunoblotting was used to validate the MS results. This analysis demonstrated that phagocytic uptake of latex beads by WT BMDMs induced the phosphorylation of DMXL1 on Ser1903, and this was blocked in $Tpl2$[D270A/D270A] BMDMs (Fig 4B). A recent study demonstrated that DMXL1, a WD40 repeat protein, interacts with the V-ATPase complex and is required for acidification of endocytic vesicles (Merkulova *et al*, 2015). Together, these results raised the interesting possibility that TPL-2 might promote phagosome acidification by inducing the phosphorylation of DMXL1.

The role of DMXL1 in intracellular vesicle acidification was previously studied in a kidney cell line by siRNA knockdown (Merkulova *et al*, 2015). Immortalised BMDMs (iBMDMs) from WT C57BL/6 and $Tpl2$[D270A/D270A] mice were used to investigate whether DMXL1 was required for phagosome maturation in macrophages. Initial experiments determined whether TPL-2 catalytic activity was required for phagosome maturation in iBMDMs similar to primary BMDMs. This was monitored following uptake of fluorescently labelled latex beads, initially by testing the effect of blocking TPL-2 catalytic activity in WT iBMDMs pharmacologically. C34 TPL-2 inhibitor significantly reduced phagosome proteolysis (Fig EV2A) and acidification (Fig EV2B). Importantly, phagosome proteolysis (Fig EV2A) and acidification (Fig EV2B) were also significantly impaired in $Tpl2$[D270A/D270A] iBMDMs relative to WT iBMDMs. These results showed that TPL-2 catalytic activity promoted phagosome maturation in immortalised macrophages, similar to primary BMDMs.

To investigate whether DMXL1 was required for phagosome acidification, $Dmxl1$ mRNA in WT iBMDMs was depleted by over 90% using $Dmxl1$ siRNA oligonucleotides (Fig 4C). Following uptake of BCECF latex beads, phagosome acidification was significantly reduced compared to WT iBMDMs transfected with non-targeting siRNAs (Fig 4D). DMXL1 depletion also decreased phagosome proteolysis monitored using DQ green BSA latex beads. (Fig EV3A). These results demonstrated that DMXL1 expression was required for phagosome acidification in macrophages. The observation that DMXL1 expression was also required for phagosome proteolysis was consistent with the known role of V-ATPase-mediated acidification for the induction of proteolytic activity in phagosomes (Lennon-Duménil *et al*, 2002; Kinchen & Ravichandran, 2008). Depletion of DMXL2, a paralog of DMXL1, did not alter phagosome acidification or proteolysis in macrophages (Fig EV3B–D).

To study whether TPL-2 and DMXL1 function in the same pathway, the effect of DMXL1 depletion on phagosome maturation was monitored in $Tpl2$[D270A/D270A] iBMDMs. DMXL1 siRNA knockdown did not alter phagosome acidification (Fig 4E) relative to transfection with a non-targeting siRNA pool. Similarly, DMXL1 depletion in $Tpl2$[D270A/D270A] iBMDMs did not alter phagosomal proteolysis compared to $Tpl2$[D270A/D270A] iBMDMs transfected with a non-targeting siRNA pool (Fig EV3E). These results were consistent with TPL-2 catalytic activity promoting phagosome acidification and proteolysis via DMXL1.

To investigate the role of TPL-2-regulated DMXL1 phosphorylation in DMXL1 function, DMXL1-depleted iBMDMs were transfected with an expression plasmid encoding a 274 amino acid DMXL1 fragment containing the two TPL-2-regulated phosphorylation sites. Expression of 3xFLAG-DMXL1$^{WT}$[1,773–2,047] partially rescued the defect in phagosome acidification induced by DMXL1 knockdown

(Fig 4F). Expression of 3xFLAG-DMXL1$^{WT}$[1,773–2,047] also partially rescued impaired phagosome proteolysis resulting from DMXL1 depletion (Fig EV3F). In contrast, expression of 3xFLAG-DMXL1$^{S1903A/S1904A}$[1,773–2,047], lacking the TPL-2 regulated phosphorylation sites, had no effect on phagosome acidification (Fig 4F) or phagosome proteolysis (Fig EV3F). Furthermore, expression

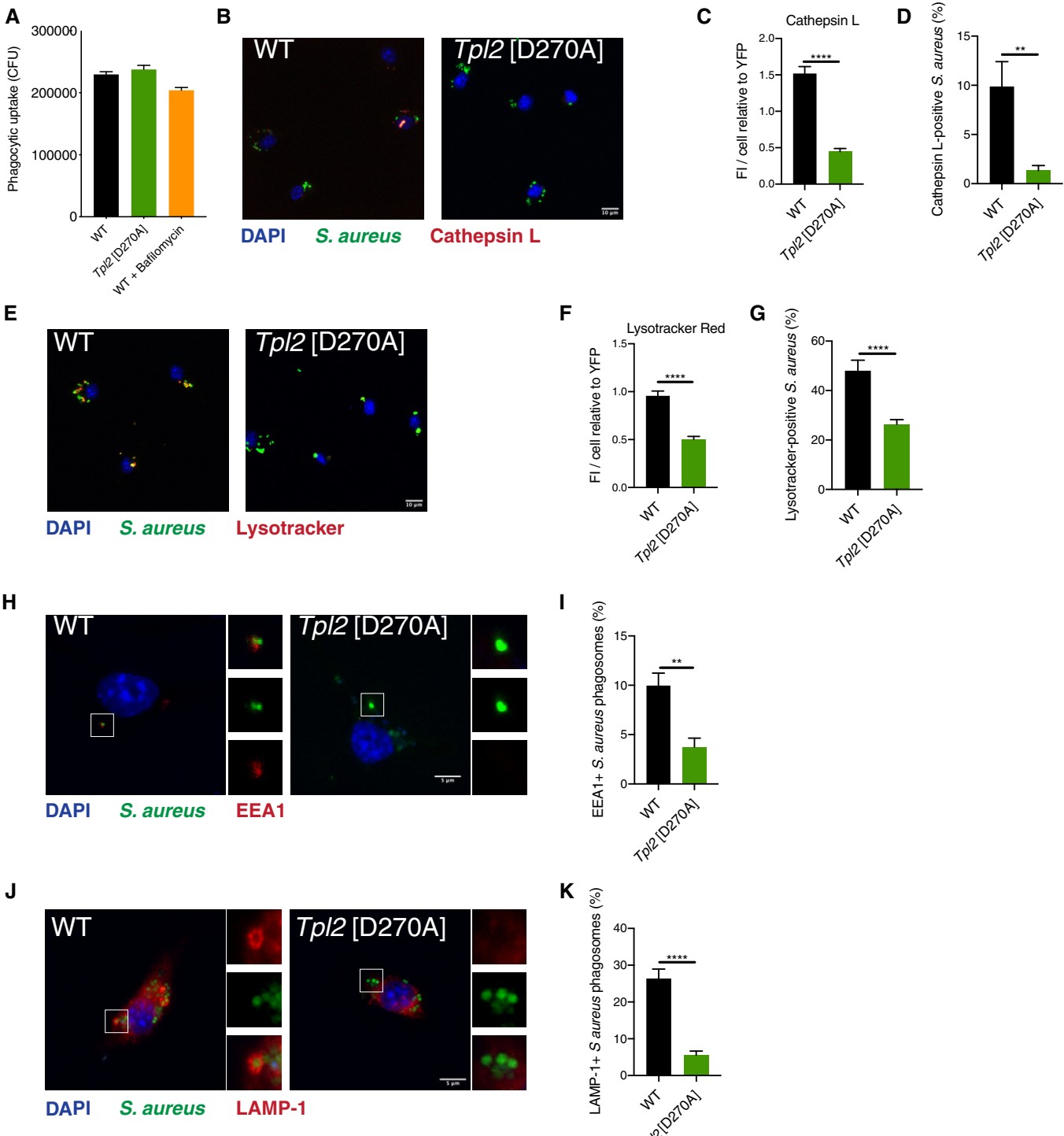

**Figure 5.**

**Figure 5.  *Tpl2*[D270A] mutation impairs maturation of *Staphylococcus aureus* phagosomes.**

A–K   BMDMs of the indicated genotypes were infected with YFP-*S. aureus* (MOI 10) for 1 h. (A) Phagocytic uptake (CFU) of YFP-labelled *S. aureus* into BMDMs of indicated genotypes 1 h post-infection ($n$ = 12; 2 biological and 6 technical replicates per condition per experiment). (B) Representative images of YFP$^+$ BMDMs labelled with Magic Red cathepsin L substrate (red) and DAPI (nuclear stain) 1 h post-infection. (C) Quantification of Magic Red cathepsin L staining from panel (A). Average fluorescence intensity of Magic Red cathepsin L substrate per cell relative to fluorescence intensity of YFP-*S. aureus* ($n$ = 26–29 cells). (D) Quantification of Magic Red cathepsin L staining co-localised with YFP-*S. aureus* from panel (A) ($n$ = 26–29 cells). (E) Representative images of YFP$^+$ BMDMs labelled with LysoTracker Red DND-99 dye (red) and DAPI, 1 h post-infection with YFP-*S. aureus* (green). (F) Quantification of LysoTracker Red signal from panel (D). Average fluorescence intensity of LysoTracker Red per cell relative to fluorescence intensity of YFP-*S. aureus* ($n$ = 66–98 cells). (G) Quantification of LysoTracker Red co-localisation with YFP-*S. aureus* from panel (D) ($n$ = 66–98 cells). (H) Representative images of BMDMs stained with an anti-EEA1 antibody (red) and DAPI, 1 h post-infection with YFP-*S. aureus* (green). (I) Quantification of EEA1$^+$ *S. aureus*-containing phagosomes from panel (G) ($n$ = 27 cells). (J) Representative images of BMDMs stained with an anti-LAMP-1 antibody (red) and DAPI, 1 h post-infection with YFP-*S. aureus* (green). (K) Quantification of LAMP-1$^+$ *S. aureus* phagosomes from panel (I) ($n$ = 25 cells).

Data information: One representative experiment out of three shown. Error bars represent SEM. $P$ < 0.01, ****$P$ < 0.0001. Student's unpaired *t*-test.

of 3xFLAG-DMXL1$^{WT}$[1,773–2,047] did not rescue phagosome acidification (Fig 4G) or phagosome proteolysis (Fig EV3G) in DMXL1-depleted *Tpl2*$^{D270A/D270A}$ iBMDMs. These findings indicate that TPL-2-mediated phosphorylation of DMXL1, and not DMXL1 expression alone, was required to drive phagosome maturation.

V-ATPase function is controlled by the reversible association of the integral membrane V0 subcomplex with peripheral membrane V1 subcomplex to form active V0/V1 holoenzyme (Kane, 2012). V0/V1 assembly is controlled by the RAVE (regulator of H$^+$-ATPase of vacuolar and endosomal membranes) complex, which includes DMXL2 (Sethi *et al*, 2010). TPL-2 regulation of phagosome acidification via DMXL1, a paralog of DMXL2 with 54% amino acid identity, raised the possibility that TPL-2 may phagosome acidification by regulating V-ATPase assembly. To investigate this, the ATP6V1C1 component of the V1 subcomplex was immunoprecipitated from lysates of *Tpl2*$^{D270A/D270A}$ and WT BMDMs following uptake of latex beads. Immunoblotting demonstrated that co-immunoprecipitation of ATP6V0D1, a component of the V0 subcomplex, was significantly decreased by *Tpl2*$^{D270A}$ mutation (Fig 4H). These results indicate that TPL-2 catalytic activity promotes V-ATPase assembly in macrophages.

Together, the results in this section suggest that TPL-2 stimulates phagosome acidification in macrophages by regulating the phosphorylation of DMXL1 to control the assembly of active V-ATPase complexes.

**TPL-2 catalytic activity induces phagosome maturation following phagocytic uptake of *Staphylococcus aureus***

Phagocytosis is a critical component of the innate immune response to bacterial infection. To investigate whether TPL-2 regulates phagosome maturation following phagocytic uptake of bacteria, BMDMs were infected with *S. aureus*. *Tpl2*$^{D270A}$ mutation did not alter the phagocytic uptake of YFP-labelled *S. aureus* (Fig 5A), consistent with results using latex beads. However, the total levels of active cathepsin per YFP$^+$ cell were significantly reduced, as detected using a labelled cathepsin L peptide substrate and confocal microscopy (Fig 5B and C). Furthermore, the percentages of cathepsin L-positive *S. aureus* phagosomes in *Tpl2*$^{D270A/D270A}$ BMDMs were decreased approximately five-fold compared to WT BMDMs (Fig 5D).

Phagosome acidification was also analysed following *S. aureus* uptake by LysoTracker Red staining and confocal microscopy. *Tpl2*$^{D270A}$ mutation decreased the fluorescence intensity for Lyso-Tracker Red per YFP$^+$ cell (Fig 5E and F). The percentage of

LysoTracker Red$^+$ phagosomes containing *S. aureus* was also significantly reduced (Fig 5G). The effect of TPL-2 catalytic activity on phagosome maturation was also investigated using specific markers phagosome maturation. Co-localisation of YFP$^+$ *S. aureus* with the early endosomal antigen 1 (EEA1), a marker of early phagosomes, was reduced in *Tpl2*$^{D270A/D270A}$ BMDMs compared to WT cells (Fig 5H and I). Recruitment of LAMP-1, a marker for late phagosomes and phagolysosomes, to YFP$^+$ *S. aureus* phagosomes was also reduced by *Tpl2*$^{D270A}$ mutation (Fig 5J and K).

Together, these results indicated that TPL-2 catalytic activity in macrophages promotes maturation of phagosomes containing internalised *S. aureus*, stimulating phagosome acidification and cathepsin activation. These results confirmed our earlier findings monitoring the effects of TPL-2 signalling on phagosome maturation in macrophages following uptake of latex beads.

**Tpl2$^{D270A}$ mutation impairs killing of phagocytosed bacteria**

Following engulfment, bacterial killing by macrophages is dependent on the maturation of phagosomes to generate phagolysosomes that have microbicidal activity. To investigate whether *Tpl2*$^{D270A}$ mutation impaired the killing of phagocytosed bacteria, BMDMs were infected with *S. aureus* and the bacterial load was measured 8h after infection by enumerating CFUs.

Bacterial killing was significantly reduced in *Tpl2*$^{D270A/D270A}$ BMDMs compared to WT cells (Fig 6A). Consistent with this, the fluorescence intensity of YFP$^+$ *S. aureus* was higher in mutant cells compared to WT controls (Fig 6B). Importantly, the degree by which *Tpl2*$^{D270A}$ mutation impaired bacterial killing was comparable to that detected by treatment of WT BMDMs with the V-ATPase inhibitor bafilomycin A1 (Fig 6A), which has pronounced inhibitory effects on phagosome maturation (Yoshimori *et al*, 1991; Bidani *et al*, 2000).

The partial inhibitory effect of bafilomycin A1 on bacterial killing implied that BMDMs also employed a phagosome-independent mechanism to kill *S. aureus*. This was likely due to the escape of bacteria from phagosomes and/or injection of pore-forming toxins into the cytosol, which result in activation of inflammasomes and induction of pyroptosis (Seilie & Wardenburg, 2017). Consistent with this, *S. aureus* infection decreased the viability of BMDMs (Fig 6C) but this was not altered by *Tpl2*$^{D270A}$ mutation, as indicated by comparable levels of released lactate dehydrogenase (Fig 6C). Furthermore, gasdermin D p30 immunoblotting demonstrated that *S. aureus* infection induced

similar levels of pyroptosis in $Tpl2^{D270A/D270A}$ and WT cells (Fig 6D). These data show that decreased bacterial killing in $Tpl2^{D270A/D270A}$ BMDMs compared to controls is not due to changes in host cell death and that pyroptosis of BMDMs induced by *S. aureus* infection is independent of TPL-2 catalytic activity.

To determine whether TPL-2 catalytic activity promoted the killing of Gram-negative bacteria, BMDMs were infected with *Citrobacter rodentium*. Phagocytic uptake of bacteria was similar between $Tpl2^{D270A/D270A}$ and WT BMDMs (Fig 6E). Measurement of the bacterial load at 2 h showed that $Tpl2^{D270A}$ mutation impaired

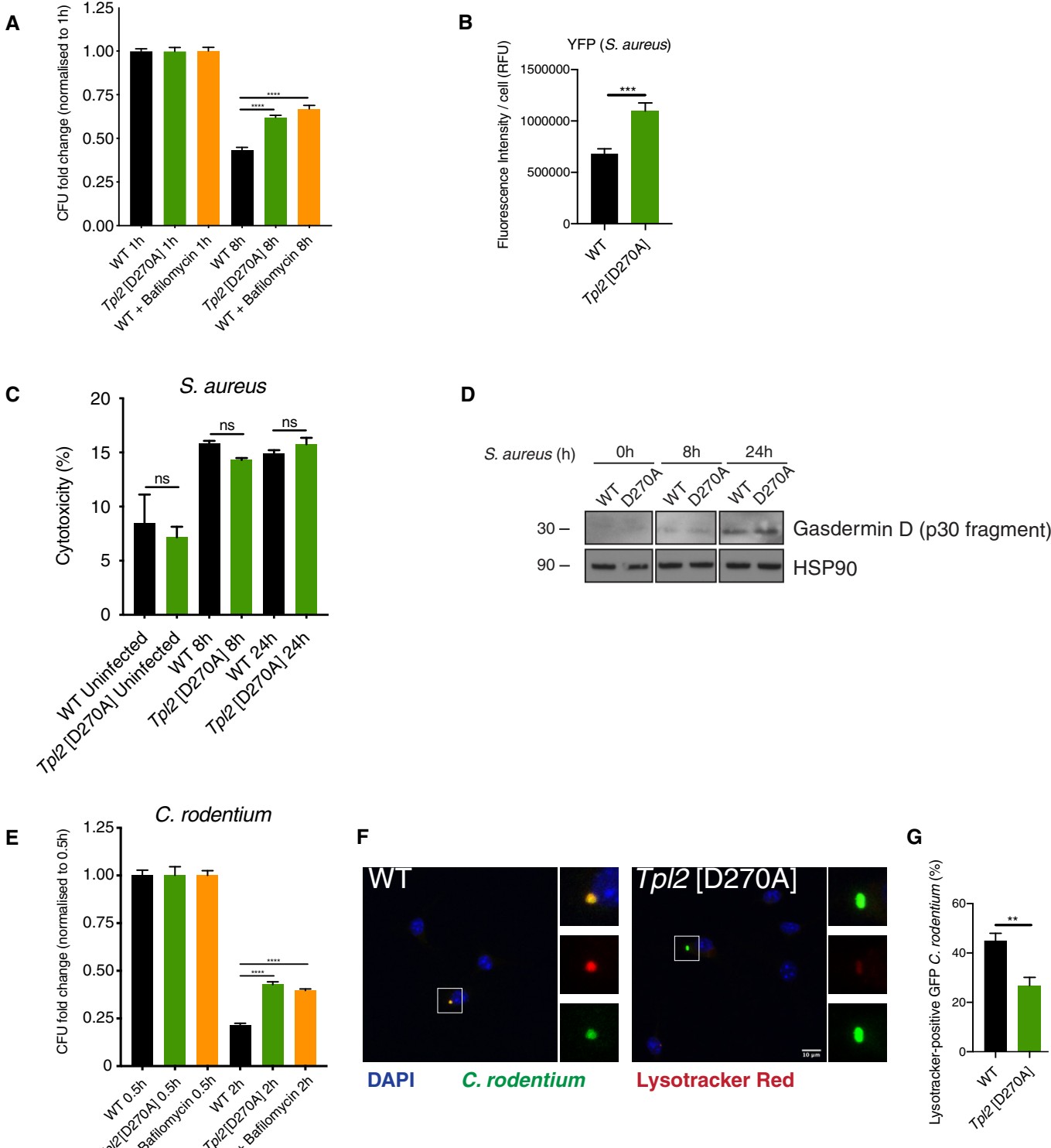

**Figure 6.**

**Figure 6. *Tpl2*[D270A] mutation impairs macrophage killing of *S. aureus*.**

A   BMDMs of the indicated genotypes were infected with YFP-*S. aureus* (MOI 10) for 0.5 h. *S. aureus* CFU were assessed at 8 h after infection and normalised to CFU at 1 h (*n* = 12; 2 biological and 6 technical replicates per condition per experiment). As a control, WT cells were treated with bafilomycin A to block V-ATPase acidification of phagosomes and phagosome proteolysis.

B   Quantification of YFP-*S. aureus* average fluorescence intensity per cell from Fig 5 panel (A) (*n* = 66–85 cells).

C   Cell death following was assayed by measuring LDH release from cytosol of BMDMs 8 and 24 h post-infection with *S. aureus* (MOI 10; *n* = 4 wells).

D   Whole cell extracts from WT and *Tpl2*[D270A] BMDMs following *S. aureus* infection (MOI 10) were immunoblotted for the gasdermin D p30 cleavage product. HSP90 was used as a loading control.

E   BMDMs of the indicated genotypes were infected with GFP-*C. rodentium* (MOI 2) for 0.5 h. *C. rodentium* CFU were assessed at 2 h after infection and normalised to CFU at 0.5 h (*n* = 12; 2 biological and 6 technical replicates per condition per experiment). As a control, WT cells were treated with bafilomycin A.

F   Representative images of GFP⁺ BMDMs labelled with LysoTracker Red DND-99 dye (red) and DAPI, 1 h post-infection with GFP-*C. rodentium* (green).

G   Quantification of LysoTracker Red co-localisation with GFP-*C. rodentium* from panel (F) (*n* = 54–63 cells).

Data information: One representative experiment out of three shown. Error bars represent SEM. $**P < 0.01$, $***P < 0.001$, $****P < 0.0001$, not significant (ns). Panels (A, B, C, E, G) one-way ANOVA.

the killing of *C. rodentium*, to a similar extent as bafilomycin A1 treatment (Fig 6E). Acidification of phagosomes containing GFP-*C. rodentium* was monitored by confocal microscopy and LysoTracker Red staining. $Tpl2^{D270A}$ mutation significantly decreased the percentage of LysoTracker Red⁺ phagosomes containing *C. rodentium* (Fig 6F and G).

TPL-2 catalytic activity is therefore required for the acidification of phagosomes containing Gram-positive or Gram-negative bacteria following infection of macrophages and for efficient killing of these bacteria.

## Discussion

Macrophage killing of phagocytosed bacteria is critical for controlling bacterial infections in the initial phase of an innate immune response. The recruitment of V-ATPases to phagosomes during their maturation is required for phagosome acidification, which is essential for further maturation to generate microbicidal phagolysosomes containing active cathepsin proteases (Kinchen & Ravichandran, 2008). In this study, genetic and pharmacological experiments demonstrated that TPL-2 is a positive regulator of phagosome acidification and proteolytic activity in mouse and human primary macrophages after uptake of latex beads. Consistent with this, following infection of macrophages by *S. aureus*, TPL-2 kinase activity stimulates acidification and activation of cathepsins in phagosomes containing internalised bacteria, promoting their efficient killing.

The stimulation of MAP kinase signalling in myeloid cells by TPL-2 during innate immune responses is well established (Gantke *et al*, 2012). TPL-2 is essential for activation of ERK1/2 MAP kinases by multiple TLRs in macrophages (Banerjee *et al*, 2006) and also contributes to TLR activation of p38α MAP kinase (Pattison *et al*, 2016). However, pharmacological and genetic experiments herein showed that TPL-2 stimulated phagosome acidification and proteolysis independently of MAP kinase activation. Phosphoproteomic analyses identified that TPL-2 catalytic activity is required for Ser1903 and Ser1904 phosphorylation of DMXL1 in macrophages following phagocytic uptake of latex beads. DMXL1 is a recently identified V-ATPase interacting protein that regulates V-ATPase-mediated acidification of endocytic vesicles in a kidney cell line (Merkulova *et al*, 2015). Knockdown experiments demonstrated that DMXL1 was also required for phagosome acidification (and

proteolysis) in macrophages after latex bead internalisation, while DMXL2 was dispensable. Expression of a DMXL1 fragment containing the Ser1903/Ser1904 phosphorylation sites rescued the acidification defect in DMXL1-deficient WT macrophages but not DMXL1-deficient $Tpl2^{D270A/D270A}$ macrophages. Together, these results indicate that TPL-2 induced phosphorylation of DMXL1 Ser1903 and Ser1904 promotes V-ATPase-mediated acidification of phagosomes.

V-ATPases consist of a peripheral membrane subcomplex called V1, which contains the sites of ATP hydrolysis, associated with an integral membrane subcomplex called V0, which encompasses the proton pore. Initial studies in yeast showed that V-ATPases are regulated by the reversible interaction of V0 and V1 subcomplexes; release of V1 into the cytosol inhibits both ATPase and proton transport activities (Parra & Kane, 1998). DMXL1 is homologous to DMXL2 (also known as Rabconnectin-3) that binds to the cytosolic V1 subcomplex of V-ATPases and promotes its association with membrane-bound V0 V-ATPase subcomplex to form an active V0/V1 holoenzyme following Notch stimulation (Nagano *et al*, 2002; Sethi *et al*, 2010). Our results show that TPL-2 catalytic activity promotes V0/V1 association following uptake of latex beads by macrophages, suggesting that TPL-2 stimulates phagosome acidification by inducing the assembly of active V-ATPase complexes via control of DMXL1 phosphorylation.

A phospho-Ser1903 DMXL1 antibody showed that the phosphorylation of DMXL1 is induced by TPL-2 following macrophage uptake of latex beads. This implies that scavenger receptors involved in phagocytosis of latex beads (Sulahian *et al*, 2008) trigger a TPL-2-dependent signalling pathway that leads to Ser1903 phosphorylation of DMXL1. In future studies, it will be important to determine whether DMXL1 Ser1903 is phosphorylated directly by TPL-2 itself or a TPL-2-regulated kinase. The failure of $Nfkb1^{SSAA}$ mutation to block TPL-2 induction of phagosome maturation raises the possibility that this pathway induces DMXL1 phosphorylation while TPL-2 still complexed with NF-κB1 p105 and ABIN-2 (Yang *et al*, 2012; Pattison *et al*, 2016). However, it cannot be ruled out that phosphorylation of p105 on sites other than those targeted by the IKK complex induces p105 proteolysis following phagocytic uptake of latex beads, releasing TPL-2 from p105 to control DMXL1 phosphorylation.

Killing of *S. aureus*, measured by CFU assay, showed that $Tpl2^{D270A}$ mutation reduced the efficiency of BMDM killing of internalised bacteria. This correlated with reduced acidification and cathepsin activation in phagosomes containing fluorescently

labelled bacteria. In addition, a decreased fraction of phagosomes containing fluorescently labelled bacteria stained positive for the early and late markers for phagosome maturation, EEA1 and LAMP-1, respectively. Together, these results showed that $Tpl2^{D270A}$ mutation impaired the maturation of phagosomes containing internalised *S. aureus* bacteria, which explained the reduced efficiency of bacterial killing in $Tpl2^{D270A/D270A}$ BMDMs compared to WT controls. TPL-2 catalytic activity was also required for acidification of *C. rodentium* phagosomes and efficient killing of *C. rodentium* suggesting that TPL-2-dependent induction of phagosome maturation is important in innate immune responses to Gram-positive and Gram-negative bacterial species.

The inhibitory effect of $Tpl2^{D270A}$ mutation on *S. aureus* killing was of a similar magnitude to that observed with WT cell treatment with bafilomycin A1, which directly inhibited V-ATPase function. Since bafilomycin A1 has pronounced inhibitory effects on phagosome maturation and endosomal trafficking (Yamamoto *et al*, 1998), these experiments showed that *in vitro* killing of *S. aureus* by BMDMs was only partially mediated by phagolysosomes. In both WT and $Tpl2^{D270A/D270A}$ BMDMs, the phagosome-independent killing of internalised bacteria was likely mediated by inflammasome activation and pyroptosis induced by escape of bacteria into the cytosol or bacterial toxins (Man *et al*, 2017). Assay of cell death (LDH release) and gasdermin D cleavage demonstrated that $Tpl2^{D270A}$ mutation did not alter pyroptosis induced by *S. aureus* infection of macrophages (Aglietti *et al*, 2016). Cleaved gasdermin D has been shown to directly kill both intracellular and extracellular bacteria (Liu *et al*, 2016). *Staphylococcus aureus* released from permeabilised cells was also exposed to bactericidal gentamicin in the culture medium. Together, our experiments show that decreased phagosome-mediated killing does not prevent *S. aureus*-induced pyroptosis and that host cell death cannot explain the effects of $Tpl2^{D270A}$ mutation on killing of *S. aureus*.

TPL-2, therefore, has two key roles in the innate immune response of macrophages to bacterial infection. TPL-2 activation of MAP kinase pathways, triggered by TLRs at the cell surface, and, independently of MAP kinase activation, TPL-2 stimulates V-ATPase-mediated acidification of phagosomes to promote more efficient bacterial killing in phagolysosomes. These processes function together to limit bacterial replication in innate immune responses.

## Materials and Methods

### Mice

Mouse strains were bred in a specific-pathogen-free environment at the Francis Crick Institute (London, United Kingdom), and all experiments carried out in accordance with regulations of the UK Home Office. $Map3k8^{D270A/D270A}$, $Nfkb1^{SSAA/SSAA}$, and $Nfkb1^{SSAA/SSAA}$ $Map3k8^{D270A/D270A}$ have been described previously (Sriskantharajah *et al*, 2009, 2014). All mouse strains were fully backcrossed on to a C57BL/6Jax background (The Jackson Laboratory).

### Reagents

PD0325901 (0.1 µM; pre-treatment for 10 min) was produced by the Division of Signal Transduction Therapy (DSTT), University of Dundee. VX-745 (1 µM; pre-treatment for 1 h) was purchased from Selleckchem. C34 (10 µM; pre-treatment for 1 h) was from MedChemExpress. Leupeptin (100 µg/ml; pre-treatment for 1 h) and bafilomycin A1 (1 µM; pre-treatment for 15 min) were purchased from Sigma-Aldrich. Lipopolysaccharide (LPS) from *Salmonella minnesota* R595 was from Enzo Life Sciences.

### Plasmids

3xFLAG-DMXL1 (1,773–2,047) and 3xFLAG-DMXL1 S1903A/S1904A (1,773–2,047) were generated by the Division of Signal Transduction Therapy (DSTT), University of Dundee.

### Antibodies

Gasdermin D antibody (#G7422) was purchased from Sigma-Aldrich. Antibodies against p-ERK1/2 T202/Y204 (#9101), ERK1/2 (#9102), p-p38 T180/Y182 (clone D3F9, #4511), p38 (#9212) and RAB5 (clone C8B1, #3547) were obtained from Cell Signaling Technology. HSP90 (clone H-114, #sc-7947), ATP6V0D1 (clone D4, #sc-393322) and ATP6V1C1 (clone H5, #sc-166848) antibodies were from Santa Cruz Biotechnology. EEA1 antibody (clone 14, #610457) was from BD Biosciences. LAMP-1 (clone 1D4B, #AB_528127) and vimentin (clone H5, #AB_528506) antibodies were from Developmental Studies Hybridoma Bank, University of Iowa. Custom-made phospho-antibody against p-DMXL-1 S1903 (Bleed 2, #DA048) was generated by the Division of Signal Transduction Therapy (DSTT), University of Dundee. All primary antibodies for Western blotting were used at 1:1,000 dilution, while primary antibodies for immunofluorescence were used at 1:200 dilution. Secondary IgG (H + L) antibodies for Western blotting were purchased from Southern Biotech and used at 1:5,000. Secondary Alexa Fluor antibodies for immunofluorescence were purchased from Thermo Fisher Scientific and used at 1:1,000.

### Mouse macrophages

Bone marrow-derived macrophages (BMDMs) were grown from femurs of 6- to 12-week-old WT and knock-in C57BL/6 mice (Kaiser *et al*, 2009). Bone marrow cells were extracted and cultured in complete medium comprising RPMI-1640 (Sigma-Aldrich) plus 10% FBS, 10 mM HEPES, 1 mM sodium pyruvate, 100 U/ml penicillin, 100 µg/ml streptomycin, 2 mM L-glutamine, 50 µM β-mercaptoethanol and 20% L929 cell supernatant. Cells were differentiated in bacteriological Sterilin dishes (Thermo Fisher Scientific). BMDMs were fed with complete medium on day 4 and harvested on day 7. Cells were seeded in complete RPMI-1640 medium minus L929 cell supernatant. BMDMs were seeded in tissue culture-treated Nunc dishes (Thermo Fisher Scientific). Immortalised BMDMs (iBMDMs) were generated by infection of BMDMs with J2 virus containing both v-raf and v-myc oncogenes following a published protocol (Gandino & Varesio, 1990). iBMDMs were cultured in complete RPMI-1640 containing 10% L929 cell supernatant.

### Human monocyte-derived macrophages

Blood from healthy donors was used in this study, which was approved by the NHS Health Research Authority and complied with

the Human Tissue Act 2004. Human peripheral blood mononuclear cells (PBMCs) were isolated from buffy coats provided by NHS Blood and Transplant Colindale. PBMCs were isolated from peripheral blood by density gradient centrifugation using Ficoll-Paque Plus (GE Healthcare). PBMCs were labelled with CD14 MicroBeads (MACS Miltenyi Biotec). CD14$^+$ PBMCs were applied to, washed and eluted from LS columns (MACS Miltenyi Biotec) according to manufacturer instructions. CD14$^+$ PBMCs were seeded at $2 \times 10^7$ cells per 140 mm Nunc dish (Sigma-Aldrich) in RPMI-1640 medium containing 10% FBS, 10 mM HEPES, 1 mM sodium pyruvate, 100 U/ml penicillin, 100 µg/ml streptomycin, 2 mM L-glutamine and 50 ng/ml human GM-CSF (STEMCELL Technologies). Cells were fed on day 4 and resulting macrophages harvested on day 7.

## Phagocytic uptake assay

Carboxylated silica beads were coupled to Alexa Fluor 488 succinimidyl ester (Thermo Fisher Scientific) (Yates & Russell, 2008). BMDMs were incubated with coupled beads at 1:300 in 5% FBS in Dulbecco's phosphate-buffered saline (DPBS) for 30 min at 37°C. External beads were removed by washing and extracellular fluorescence quenched using trypan blue. Cell-associated fluorescence intensities were measured in a CLARIOstar (BMG Labtech) microplate reader at excitation/emission wavelengths 490/520 nm.

## Phagosome proteolysis assay

Bead coupling procedures and proteolysis assays were adapted from protocols developed by the Valenzeno and Russell laboratories (Russell *et al*, 1995; Yates *et al*, 2005; Yates & Russell, 2008). DQ Green BSA (Thermo Fisher Scientific) and Alexa Fluor 594 succinimidyl ester (Thermo Fisher Scientific) were coupled to carboxylated 3 µm silica beads (Kisker Biotech), as described (Yates *et al*, 2005; Yates & Russell, 2008). BMDMs were seeded on 96-well plates (Greiner CELLSTAR®) at $1 \times 10^5$ cells/well and cultured overnight. DQ Green BSA/Alexa Fluor 594-coupled silica beads were diluted 1:300 in 5% FBS in DPBS and incubated with cells for 3 min at RT. External beads were removed by washing with pre-warmed 5% FBS in DPBS, and real-time fluorescence intensities of cells were measured at ex/em 490/520 nm and at ex/em 590/620 nm at 37°C using a CLARIOstar (BMG Labtech) reader at 2min intervals over 6 h. The calibration fluorophore Alexa Fluor 594 was used to normalise against phagocytic bead uptake. Plots were generated from DQ Green BSA to Alexa Fluor 594 ratios.

## Phagosome acidification assay

Silica beads were coupled to BCECF acid (2′,7′-Bis-(2-Carboxyethyl)-5-(and-6)-Carboxyfluorescein) (Thermo Fisher Scientific) (Russell *et al*, 1995). BCECF-coupled beads were diluted 1:200 in DPBS + 5% FBS and incubated with cells for 3 min at RT. After washing to remove external beads, real-time fluorescence intensities were measured at ex/em 485/520 nm at 37°C using a PHERAstar Plus (BMG Labtech) reader at 1 min intervals over 1 h. BCECF is a pH indicator with optimal fluorescence at pH 7. As phagosomal pH rapidly decreases, fluorescence intensity of the bead-coupled pH indicator BCECF decreases at 485nm/520nm.

## Oxidative burst assay

OxyBURST Green H2HFF BSA (Thermo Fisher Scientific) and Alexa Fluor 594 succinimidyl ester (Thermo Fisher Scientific) were coupled to carboxylated 3 µm silica beads (Kisker Biotech), as described (Vanderven *et al,* 2009). OxyBURST Green BSA/Alexa Fluor 594-coupled silica beads were diluted 1:200 in 5% FBS in DPBS and incubated with BMDMs for 3 min at RT. External beads were removed by washing and replaced with pre-warmed 5% FBS in DPBS. Real-time fluorescence intensities of cells were measured at ex/em 480/520 nm and at ex/em 590/620 nm at 37°C using a CLARIOstar (BMG Labtech) reader at 2 min intervals over 90 min. Oxidation of H2HFF-OxyBURST generates a fluorescein-based product, which emits a signal at 520 nm upon excitation at 480 nm (Vanderven *et al,* 2009).

## Phagosome isolation

Phagosomes were isolated from BMDMs as previously described (Trost *et al,* 2009; Dill *et al,* 2015) Briefly, 0.93 µM blue-dyed carboxylated polystyrene beads (Estapor Merck Chimie SAS) were diluted 1:50 in complete RPMI-1640 medium (without L929 cell supernatant) and induced for 30 min with $1 \times 10^7$ BMDMs per 10 cm Nunc dish. BMDMs were washed and lysed in hypotonic buffer by Dounce homogenisation. Nuclei and cell debris were removed by centrifugation 1,000 ×*g* for 5 min at 4°C. Phagosomes were then isolated using a discontinuous sucrose gradient. Phagosome fractions were separated in a SW41Ti swinging bucket rotor in an Optima™ L-90 K ultracentrifuge (Beckman Coulter) at 72,300 ×*g* for 1 h at 4°C. Blue-coloured latex bead-containing phagosomes were collected from the interface of the 10 and 25% sucrose fractions.

## Proteomic sample preparation of phagosomes

Isolated phagosomes were lysed in 5% SDS, 50 mM triethylammonium bicarbonate (TEAB) pH 7.5 (S-Trap), sonicated and protein concentration determined using the BCA Protein Assay Kit (Pierce Protein). Samples were reduced in 10 mM tris(2-carboxyethyl)phosphine for 30 min at RT followed by alkylation in 10 mM iodoacetamide for 30 min. Samples were acidified and digested on an S-trap spin column by addition of porcine trypsin at 1:10 (Pierce) for 2 h at 47°C. Peptides were eluted in 50 mM TEAB pH 8.0, 0.2% formic acid and 0.2% formic acid in 50% acetonitrile, respectively.

## Label-free MS acquisition of phagosome proteomes

Peptide samples were separated on an Ultimate 3000 RSLC system (Thermo Scientific) with a C18 PepMap, serving as a trapping column (2 cm × 100 µm ID, PepMap C18, 5 µm particles, 100 Å pore size) followed by a 50 cm EASY-Spray column (50 cm × 75 µm ID, PepMap C18, 2 µm particles, 100 Å pore size) (Thermo Scientific). Buffer A contained 0.1% formic acid and Buffer B 80% acetonitrile, 0.1% formic acid. Peptides were separated with a linear gradient of 1–35% (Buffer B) over 120 min followed by a step from 35 to 90% acetonitrile, 0.1% formic acid in 0.5 min at 300 nl/min and held at 90% for 4 min. The gradient was then decreased to 1% Buffer B in 0.5 min at 300 nl/min for 10 min. Mass

spectrometric identification was performed on an Orbitrap QE HF mass spectrometer (Thermo Scientific) operated "TopN" data-dependent mode in positive ion mode. FullScan spectra were acquired in a range from 400 $m/z$ to 1,500 $m/z$, at a resolution of 120,000 (at 200 $m/z$), with an automated gain control (AGC) of $1 \times 10^6$ and a maximum injection time of 50 ms. Charge state screening is enabled to exclude precursors with a charge state of 1. For MS/MS fragmentation, the minimum AGC was set to 5,000 and the most intense precursor ions were isolated with a quadrupole mass filter width of 1.6 $m/z$ and 0.5 $m/z$ offset. Precursors were subjected to higher-energy collisional dissociation (HCD) fragmentation that was performed in one-step collision energy of 25%. MS/MS fragments ions were analysed in the Orbitrap mass analyser with a 15,000 resolution at 200 $m/z$.

## Sample preparation for Tandem Mass Tag (TMT) phosphoproteome analysis of BMDMs

BMDMs from WT or $Tpl2^{D270A/D270A}$ mice were incubated with carboxylated polystyrene beads as for phagosome isolations. Protein content was estimated by protein assay, and a 200 μg aliquot of each sample was taken for further processing. Samples were reduced with 10 mM dithiothreitol for 25 min at 56°C and then alkylated with 20 mM iodoacetamide for 30 min at RT. The alkylation reaction was quenched with an additional 10 mM dithiothreitol, and then, each sample was diluted with 50 mM HEPES to reduce the urea concentration to < 2 M prior to digestion. Proteolytic digestion was carried out by the addition of 4 μg LysC (FUJIFILM Wako Pure Chemical Corporation) and incubated at 37°C for 2.5 h followed by the addition of 10 μg trypsin (Pierce) and overnight incubation at 37°C. After acidification, C18 MacroSpin columns (Nest Group) were used to clean up the digested peptide solutions and the eluted peptides dried by vacuum centrifugation. Samples were resuspended in 50 mM HEPES and labelled using the 0.8 mg Tandem Mass Tag 10plex isobaric reagent kit (Thermo Scientific) resuspended in acetonitrile. Labelling reactions were quenched with hydroxylamine, and a pool was made containing all of the samples. Acetonitrile content was removed from the pooled TMT solution by vacuum centrifugation and then acidified before using a Sep-Pak C18 (Waters) to clean up the labelled peptide pool prior to phospho-peptide enrichment. The eluted TMT-labelled peptides were dried by vacuum centrifugation, and phosphopeptide enrichment was subsequently carried out using the sequential metal oxide affinity chromatography (SMOAC) strategy with High Select TiO2 and Fe-NTA enrichment kits (Thermo Scientific). Eluates were immediately dried by vacuum centrifugation after enrichment and then combined prior to fractionation with the Pierce High pH Reversed-Phase Peptide Fractionation Kit (Thermo Scientific). All fractions were dried by vacuum centrifugation and stored at −80°C until the day of MS analysis.

## MS data acquisition and processing for TMT phosphoproteome

Dried TMT-labelled phosphopeptide fractions were resuspended in 0.1% trifluoroacetic acid and then analysed by online nanoflow LC-MS/MS using an Orbitrap Fusion Lumos mass spectrometer (Thermo Scientific) coupled to an Ultimate 3000 RSLCnano (Thermo Scientific). 15 μl of sample was loaded via autosampler into a 20 μl

sample loop and pre-concentrated onto an Acclaim PepMap 100 75 μm × 2 cm nanoviper trap column with loading buffer, 2% v/v acetonitrile, 0.05% v/v trifluoroacetic acid, 97.95% water (Optima grade, Fisher Scientific) at a flow rate of 7 μl/min for 6 min in the column oven held at 40°C. Peptides were gradient eluted onto a C18 75 μm × 50 cm, 2 μm particle size, 100 Å pore size, reversed-phase EASY-Spray analytical column (Thermo Scientific) at a flow rate of 275 nl/min and with the column temperature held at 40°C, and a spray voltage of 2,100 v using the EASY-Spray Source (Thermo Scientific). Gradient elution buffers were A 0.1% v/v formic acid, 5% v/v DMSO, 94.9% v/v water and B 0.1% v/v formic acid, 5% v/v DMSO, 20% v/v water, 74.9% v/v acetonitrile (all Optima grade, Fisher Scientific aside from DMSO, Honeywell Research Chemicals). The gradient elution profile used was 2% B to 25% B over 120 min, increasing to 40% B over the following 25 min; then, the column was washed 90% B and re-equilibrated to 2% B to complete the 180 min run. Two replicate injections were made for each fraction with different fragmentation methods based on the MS2 HCD and MSA SPS MS3 strategies described (Jiang *et al*, 2017). The acquired raw mass spectrometric data were processed in MaxQuant (Cox & Mann, 2008) (version 1.6.2.10) for peptide and protein identification; the database search was performed using the Andromeda search engine against the Mus Musculus reference proteome canonical sequences from UniProtKB (UP000000589 download August 2017). Fixed modifications were set as Carbamidomethyl (C) and variable modifications set as Oxidation (M) and Phospho (STY). The estimated false discovery rate was set to 1% at the peptide, protein and site levels. A maximum of two missed cleavages were allowed. Reporter ion MS2 or Reporter ion MS3 was appropriately selected for each raw file. Other parameters were used as preset in the software. The MaxQuant output file PhosphoSTY Sites.txt, an FDR-controlled site-based table compiled by MaxQuant from the relevant information about the identified peptides, was imported into Perseus (v1.4.0.2) for data evaluation.

## Immunofluorescence

BMDMs were seeded on 13 × 1.5 mm round coverslips (VWR International) at $1 \times 10^5$ cells in a 24-well plate and rested overnight. Cells were treated as described in figure legends. After treatments, BMDMs were washed in PBS and fixed in eBioscience™ IC fixation buffer (Thermo Fisher Scientific) for 10 min at RT. BMDMs were permeabilised in 0.1% Triton X-100 in PBS for 4 min, washed and then blocked in 0.4% gelatin from cold water fish skin (Sigma-Aldrich) in PBS for 10 min. Following incubation in 0.1% sodium borohydride pH 8.0 (Sigma-Aldrich) for 10 min, cells were stained with primary antibody solution (1:200) for 1 h, followed by three washes. BMDMs were incubated in secondary antibody solution (1:1,000) for 30 min. Nuclei were stained by incubation with 4′,6-diamidino-2-phenylindole (DAPI) (1:1,000) for 15 min. Washed coverslips were mounted on Corning microscope slides (Merck) using Fluoromount mounting solution (Sigma-Aldrich). Slides were visualised under a Zeiss Inverted LSM 880 Axio Observer confocal laser scanning microscope (ZEISS) using a ZEISS 40× Plan-Apochromat (numerical aperture = 1.3) DIC M27 oil immersion objective. Where indicated, a ZEISS 63x Plan-Apochromat (numerical aperture = 1.4) DIC UV-VIS-IR M27 oil immersion objective was used.

## Cathepsin activity assay

Following treatment, BMDMs cultured on coverslips (as described above) were washed twice with DPBS and then incubated with the Magic Red cathepsin L substrate (ICT942, Bio-Rad Antibodies) in 5% FBS in DPBS at a final dilution of 1:250 for 1 h at 37°C. Following washing, BMDMs were fixed, stained with DAPI and imaged as detailed above. Fluorescence of the Magic Red cathepsin L substrate was monitored at ex/em 592/628 nm.

## LysoTracker red

Following treatment, BMDMs cultured on coverslips were washed twice with DPBS and then incubated with LysoTracker Red DND-99 (100nM final concentration; L7528, Thermo Fisher Scientific) in DPBS + 5% FBS for 1 h at 37°C. The immunostaining protocol was performed as described above, and fluorescence of LysoTracker Red was monitored at ex/em 577/590 nm. LysoTracker Red is an acidotropic red-fluorescent-dye, which stains acidic cellular compartments.

## ROS assay

Following treatment, BMDMs cultured on coverslips were washed twice with DPBS and then incubated with ROS Deep Red dye (100 nM final concentration; ab186029, Abcam) in DPBS + 5% FBS for 1 h at 37°C. The immunostaining protocol was performed as described above, and fluorescence of ROS Deep Red dye was monitored at ex/em 650/675 nm. ROS Deep Red is a cell permeable Deep Red-fluorescent dye, which reacts with superoxides and hydroxyl radicals.

## Image analysis

Images were analysed using ImageJ (Fiji) version 2.0.0 (National Institutes of Health, USA). Fluorescence intensities per BMDM were quantified by creating masks of cell outlines and determining the integrated density (IntDen). Ratios of integrated densities for two different markers in the same masked area were used to determine relative fluorescence intensity. Coloc 2, a Fiji plugin, was used to quantify co-localisation. To calculate Pearson's correlation coefficients and Manders' coefficients, Coloc 2 parameters were set to bisectional threshold regression with a point spread function of 3.0 and Costes randomisations of 100. Coefficients above the Coloc 2 autothreshold were selected for data analysis. For LAMP-1 staining, *S. aureus* bacteria were automatically counted by normalising the YFP fluorescence threshold across all fields of view and counting particles > 0.2 μm. Percentages of YFP-*S. aureus* associated with LAMP-1 rings were determined by analysing > 100 BMDMs including > 600 *S. aureus* bacteria per genotype from at least 40 random fields of view in three independent experiments.

## siRNA-mediated knockdown of *Dmxl1*

iBMDMs were seeded at $5 \times 10^4$ cells/well in a tissue culture-treated Nunc 24-well plate and at $1 \times 10^4$ cells/well in a 96-well plate (Greiner CELLSTAR®) in complete RPMI-1640 containing 10% L929 cell supernatant and cultured overnight at 37°C. All siRNA stocks were reconstituted in siRNA buffer (Horizon Discovery) to 20 μM.

SMARTpool ON-TARGETplus siRNA was used for *Dmxl1* (L-055471-01-0005, Horizon Discover) alongside an ON-TARGETplus non-targeting pool as siRNA control (D-001810-10-20, Horizon Discovery). Cells were transfected with siRNAs using Viromer Green (Lipocalyx) according to manufacturer instructions. Macrophages were transfected with siRNA at a final concentration of 50 nM. After 48 h culture, 96-well plates were used for bead assays, while 24-well plates were used for qRT–PCR analysis to confirm *Dmxl1* mRNA knockdown. In *Dmxl1* rescue experiments, *Dmxl1* siRNA was co-transfected with plasmids expressing either 3xFLAG-DMXL1 (1773-2047) or 3xFLAG-DMXL1 S1903A/S1904A (1,773–2,047). Each well in a 24-well and 96-well plate was co-transfected with 125 ng and 50 ng cDNA, respectively.

## Quantitative real-time PCR (qRT–PCR)

Following siRNA knockdown, iBMDMs were lysed in 350 μl Qiagen RLT lysis buffer (Thermo Fisher Scientific) containing β-mercaptoethanol at 1:100 (v/v). RNA was isolated using the Qiagen RNeasy Mini Kit (Thermo Fisher Scientific) and Qiagen RNase-free DNase set (Thermo Fisher Scientific), following manufacturer instructions. Isolated RNA was reverse transcribed to cDNA using the SuperScript VILO cDNA Synthesis Kit (Invitrogen). qRT–PCR was prepared in MicroAmp Optical 384-well plates (Applied Biosystems) with a total reaction volume of 7 μl, which contained 2 μl cDNA, 3.5 μl TaqMan Universal PCR Master Mix (Applied Biosystems), 0.35 μl TaqMan FAM-coupled probe and 1.15 μl nuclease-free water. TaqMan FAM-MGB probe Mm01261785_m1 was used for amplification of *Dmxl1* (Thermo Fisher Scientific). qRT–PCR was performed on a QuantStudio 3 and 5 (Applied Biosystems) set to comparative $C_T$ mode, and *Hprt* (Mm01545399_m1) was used as a housekeeping gene for expression level normalisation.

## SDS–PAGE and immunoblotting

Macrophages were lysed in 50 mM Tris–HCl pH 7.5, 150 mM sodium chloride, 1% Triton X-100, 10 mM sodium fluoride, 1 mM sodium pyrophosphate, 10 mM β-glycerophosphate, 2 mM EDTA, 100 μM sodium orthovanadate, 10% glycerol, supplemented with one EDTA-free protease inhibitor cocktail tablet (Roche). Protein concentration in cleared cell extracts was quantified using the Coomassie Protein Assay Kit (Pierce). Lysates were mixed with 5X sample buffer (250 mM Tris–HCl pH 6.8, 32.5% glycerol, 5% SDS, 5% β-mercaptoethanol) and heated for 10 min at 90°C. Samples were subjected to SDS–PAGE on a 10% acrylamide gel.

After electrophoresis, proteins were transferred onto PVDF membranes (Bio-Rad Laboratories) using the Blot Turbo Transfer System (Bio-Rad Laboratories). Membranes were blocked with 5% (w/v) milk powder in PBS plus 0.05% Tween-20 for 1 h. Membranes were incubated with primary antibodies overnight at 4°C, washed four times in PBS plus 0.05% Tween-20 and then incubated with horseradish-peroxidase (HRP)-conjugated secondary antibodies. After four PBS plus 0.05% Tween-20 washes, membranes were incubated with Amersham ECL Western Blotting Detection Reagent (GE Healthcare Life Sciences) or Immobilon Western Chemiluminescent HRP Substrate (Millipore) to visualise protein levels following exposure of X-ray film (Scientific Laboratory Supplies).

### V-ATPase co-immunoprecipitation

IgG Sepharose resin (Sigma-Aldrich, 15 μg packed beads) was incubated with 2.5 mg cell lysate and 2 μg/ml ATP6V1C1 antibody for 16 h at 4°C. Beads were washed four times with lysis buffer and once with sterile water to remove any buffering capacity. Beads were dried with a flat gel-loading tip. Immunoprecipitated proteins were released by glycine elution. Briefly, beads were incubated with glycine elution buffer at 1,400 rpm for 3 min, centrifuged at 14,000 *g* for 1 min at RT, and the eluate was transferred to a new Eppendorf tube. Elution was repeated three times. To neutralise the acidic pH, 1 M Tris base pH 8 was added at a final concentration of 20% (v/v). Eluates were mixed with 5× sample buffer and heated for 10 min at 90°C. Samples were subjected to SDS–PAGE.

### Bacterial colony-forming unit (CFU) assay

BMDMs were seeded in antibiotic-free RPMI-1640 medium (Sigma-Aldrich) containing 10% FBS, 10 mM HEPES, 1 mM sodium pyruvate and 50 μM β-mercaptoethanol. YFP-*Staphylococcus aureus* (*S. aureus* strain RN6390) was cultured in Luria-Bertani medium supplemented with 10 μg/ml chloramphenicol (Sigma-Aldrich) for 16 h at 37°C with constant shaking at 200 rpm. GFP-*Citrobacter rodentium* (*C. rodentium* strain ICC683) was cultured in Luria-Bertani medium supplemented with 50 μg/ml nalidixic acid (Sigma-Aldrich) and 100 μg/ml chloramphenicol (Sigma-Aldrich). *Staphylococcus aureus* culture was diluted in antibiotic-free RPMI-1640 medium and added to BMDMs at an MOI of 10:1, while *C. rodentium* culture was added at an MOI of 2:1. Plates were centrifuged at 100× *g* for 5 min at RT to facilitate phagocytic uptake.

Following bacterial infection, BMDMs were incubated for 25 min at 37°C to allow phagocytosis. BMDMs were washed in pre-warmed PBS and incubated with RPMI-1640 medium containing 50 μg/ml gentamicin (Sigma-Aldrich) up to 1 h at 37°C. After 1 h, medium was replaced with RPMI-1640 medium containing 10 μg/ml gentamicin to block division of extracellular bacteria. After further culture for 7 h, BMDMs were washed in ice-cold PBS and lysed in 1 ml 0.1% Triton X-100 in PBS for 5 min. For *C. rodentium*, phagocytic uptake was normalised to a 0.5 h time point and bacterial killing was monitored at 2 h. Lysates were diluted in PBS, plated on antibiotic-free agar plates (The Francis Crick Institute) and incubated at 37°C overnight. Colony-forming units were automatically counted using the Synbiosis aCOLyte (7510/SYN).

### Cytotoxicity assay

BMDMs of indicated genotypes were seeded in 24-well Nunc plates and infected with *S. aureus* (MOI of 10:1) for the indicated times in antibiotic-free RPMI-1640 medium containing 10% FBS, 10 mM HEPES, 1 mM sodium pyruvate and 50 μM β-mercaptoethanol. Supernatants were collected, centrifuged at 10,000 *g* for 2 min at RT to remove cell debris. As MAX control, an uninfected 24-well plate was transferred to −80°C for complete cell lysis. The Cytotoxicity Detection Kit (11 644 793 001, Roche) was used according to manufacturer instructions. Briefly, 100 μl

supernatant was combined with 100 μl substrate for 10 min at RT protected from light. 100 μl 0.16 M sulphuric acid was added to terminate the assay. LDH (lactate dehydrogenase) activities in supernatants were measured at 490 nm, with a reference wavelength set to 650 nm.

### Statistical analysis

All data analyses were performed using GraphPad Prism software (version 8.2.1). Unless otherwise stated in figure legends, data were compared using a two-way ANOVA. Error bars represent standard errors of the mean (SEM). Statistical significance of data is represented by asterisks (*) corresponding to *P* value thresholds: not significant (ns) $P > 0.05$, *$P < 0.05$, **$P < 0.01$, ***$P < 0.001$, ****$P < 0.0001$.

## Data availability

The mass spectrometry phagosome proteomics data have been deposited to the ProteomeXchange (Deutsch *et al*, 2017) Consortium via the PRIDE (www.ebi.ac.uk/pride) (Perez-Riverol *et al*, 2019) partner repository, with the dataset identifier PXD020401. The phosphoproteomics dataset identifier is PXD020247.

Expanded View for this article is available online.

### Acknowledgements

We are grateful to The Francis Crick Institute Biological Research, Advanced Light Microscopy and Media Preparation Science Technology Platforms for their help over the course of this work. We also thank Professor David W. Holden (Imperial College London) and Dr. Max Gutierrez (The Francis Crick Institute) for advice on bacterial infections, Professor Tracy Palmer (Newcastle University) for providing the YFP-*S. aureus* strain, Professor Gad Frankel (Imperial College London) for providing the GFP-*C. rodentium* strain, Dr José Luis Marin Rubio (Newcastle University) for assistance with phagosome isolations, Dr Antonio Tedeschi and Dr Jeremy Carlton (The Francis Crick Institute) for advice about vesicle trafficking and Louise Blair as well as other members of the Ley Laboratory for technical advice and discussions throughout these studies. This work was supported by The Francis Crick Institute, which is funded by the Medical Research Council, Cancer Research UK and the Wellcome Trust. Felix Breyer was funded on a Boehringer Ingelheim Fonds PhD Fellowship. Matthias Trost was funded through a Wellcome Trust Investigator Award (215542/Z/19/Z).

### Author contributions

FB performed the majority of experiments. AH helped establish phagosome bead assays. TT provided advice about *S. aureus* infection of macrophages and CFU measurement. AH and TH performed phagosome isolations together with FB. HRF assisted with sample preparation for phosphoproteome mass spectrometry. HRF and APS co-designed and performed phosphoproteome mass spectrometry experiments. PC and JP performed data analysis for phagosome proteomics. JJ managed and maintained mouse colonies. MT provided advice for the project and edited the manuscript. FB and SCL designed the experiments and wrote the manuscript.

### Conflict of interest

The authors declare that they have no conflict of interest.

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
