## [Review Process File · The EMBO Journal]

TPL-2 kinase induces phagosome acidification to promote macrophage killing of bacteria

Felix Breyer, Anetta Härtlova, Teresa Thurston, Helen Flynn, Probir Chakravarty, Julia Janzen, Julien Peltier, Tiaan Heunis, Ambrosius P. Snijders, Matthias Trost and Steven Ley

DOI: [10.15252/embj.2020106188](https://doi.org/10.15252/embj.2020106188)

Corresponding author(s): STEVEN LEY (sley@ic.ac.uk)

Review Timeline:

Submission Date:	8th Jul 20
Editorial Decision:	6th Aug 20
Revision Received:	11th Feb 21
Editorial Decision:	5th Mar 21
Revision Received:	11th Mar 21
Accepted:	15th Mar 21

Editor: Karin Dumstrei

Transaction Report:

Dear Steven,

Thank you for submitting your manuscript to The EMBO Journal. Your study has now been seen by two referees and their comments are provide below.

As you can see from the comments the referees appreciate the analysis but also find that much further insight would be needed to consider publication here. Should you be able to address the raised concerns in a good way then I am open to consider a revised version. I should point out that I need support from referees to move forward with the manuscript. I am happy to discuss the raised points further and maybe it would be most helpful to do so via phone or video.

We generally allow three months as standard revision time. As a matter of policy, competing manuscripts published during this period will not negatively impact on our assessment of the conceptual advance presented by your study. Should you foresee a problem in meeting this three-month deadline, please let me know in advance and I can grant an extension.

Thank you for the opportunity to consider your work for publication. Looking forward to discussing the revisions further with you

Yours sincerely,

Karin

Karin Dumstrei, PhD
Senior Editor
The EMBO Journal

When assembling figures, please refer to our figure preparation guideline in order to ensure proper formatting and readability in print as well as on screen:
<http://bit.ly/EMBOPressFigurePreparationGuideline>

Further information is available in our Guide For Authors:

The revision must be submitted online within 90 days; please click on the link below to submit the revision online before 4th Nov 2020.

Referee #1:

In this manuscript, Breyer et al report the role of the kinase TPL-2 in the regulation of phagosome maturation and the killing of *S. aureus*, in macrophages. Mechanistically, the authors show that this TPL-2 function is independent of ERK1/2 and p38 activation, and that TPL-2 inhibition reduces V-ATPase in the phagosomes and the phosphorylation of its regulator, DMXL1. Overall, the manuscript is well written. The finding that TPL-2 controls phagosome acidification and proteolysis independently of MAPK activation is novel and may be potentially interesting. Although most experiments in this study were appropriately designed and executed, the authors need to provide more data to substantiate the manuscript. Overall, the study is preliminary.

Major concerns:

Does TPL-2 regulate phagosome maturation and the killing in other phagocytic cells such as neutrophils or dendritic cells? Is the phagocytosis and the killing of other microorganisms (gram-negative, fungus) also controlled by TPL-2?

Is DMXL1 directly phosphorylated by TPL-2? If not, which kinase is phosphorylating DMXL1? Is DMXL1 phosphorylation on S1903 and S1904 modulating V-ATPase recruitment to the phagosome, or V-ATPase activity, or both? How does TPL-2 regulate interactions between V-ATPase and DMXL1? Is DMXL1 phosphorylation on S1903 and S1904 modulating its interaction with V-ATPase?

How are acidification and proteolysis affected in TPL2D270A cells after knocking down DMXL1? And after expressing the 3xFLAG-DMXL1WT? Does 3xFLAG-DMXL1WT also rescue in TPL2D270A cells? What is the effect of phospho-mimicking peptide 3xFLAG-DMXL1SSEE or 3xFLAG-DMXL1SSDD in both TPL2D270A and WT cells? Are the DMXL1 peptides (WT and mutants) interacting with V-ATPase? If so, how does this interaction regulate the ATPase activity? How are the acidification and the killing of *S. aureus* in WT cells expressing 3xFLAG-DMXL1WT or 3xFLAG-DMXL1SSAA or 3xFLAG-DMXL1SSDD affected? and in the TPL2D270A cells?

Experimental conditions in Figure 1B are not clear. It is the leupeptin included all the time in the assay, and in all conditions?

Many of the experiments in this work are performed using fluorescence beads. How do the authors know that they have cleaned all the external beads attached to the cell membrane? The same question when the authors perform experiments using YFP-*S. aureus*. This is relevant for the interpretation of the phagocytic uptake results and relevant controls should be shown.

The way the authors represent proteolysis and acidification data is not clear, especially in Fig 2. The colours chosen to represent different experimental conditions are similar and this make very difficult to distinguish from each other. For example: in Fig. 2A and 2B, WT and Nfkb1SSAA data are not clear. In Fig 2D, the WT data cannot be seen. In Fig 2E, the Nfkb1SSAA data cannot be seen.

In Fig 3, a control showing protein that do not change is missing. Is the decrease in proteins in TPL2D270A samples real? Does TPL2D270A cells have less phagosomes?

If the killing of *S.aureus* is diminished in TPL2D270A why the cytotoxicity is the same than in WT?

It is not clear what does the "n" represent in each figure (field/ cells / phagosomes?). The statistical comparison of data in all figures are not clear either.

Minor concerns:

Are ERK1/2 and p38 pathways activated after macrophage uptake of latex beads?

In Fig. 2, the authors should provide blots of total ERK1/2 or p38.

If TPL-2 controls p38 phosphorylation, why this is not reduced in TPL2KD/Nfkb1SSAA BMDM stimulated with LPS?

On page 6 the authors mention: "Nfkb1SSAA mutation, which changes both serines to alanine, blocks IKK-induced p105 proteolysis (Figure 2)". p105 proteolysis is not shown in Fig. 2.

On page 10: "Consistent with this, *S. aureus* infection decreased the viability of BMDMs (Figure 6B)" it should be (Figure 6C)

What does US, UT and NT mean in Fig 4C and F

Other suggestions:

It would be interesting to see if incubation with Bafilomycin of TPL2D270A cells completely revert

the effect of TPL2 mutation (Fig 6A).

Referee #2:

The study by Breyer and colleagues contains an impressive combination of a variety of techniques to demonstrate that phagosome acidification is strongly dependent on the ability of the TPL-2 kinase to phosphorylate proteins, most notable DMXL1, which is known to associate with the V-ATPase complex. The study has been performed with considerable rigor and the data are compelling and for the most part support the major conclusions made by the authors. I have two points that if addressed would add to the paper.

1. The authors show the reduced acidification through the signal reduction upon protonation of carboxyfluorescein that is coupled to reporter particles. These data are very clear and convincing. However, for ease of analysis and to allow comparison between this study and published studies it would be helpful if the authors converted these into real pH values. This can be achieved using pH standards as detailed in the reference Yates and Russell, 2008, which is already included in the manuscript.

2. The authors show that the generation of reactive oxygen intermediate by the NADPH oxidase complex is reduced, Figure 1F. The link between acidification, phagosome maturation and NADPH oxidase activity is not fully resolved. It is possible that TPL-2 kinase has a direct impact on the formation of the NADPH oxidase complex, do the authors have any data on this? Does blocking acidification with Bafilomycin A impact NADPH oxidase activity? How does this compare with inhibition with C34? It would be of value for the authors to utilize reporter beads for the NADPH oxidase activity to generate more quantitative kinetic data, as documented in VanderVen, Yates and Russell, 2009, Traffic 10: 372.

REVIEWER #1:

In this manuscript, Breyer et al report the role of the kinase TPL-2 in the regulation of phagosome maturation and the killing of S. aureus, in macrophages. Mechanistically, the authors show that this TPL-2 function is independent of ERK1/2 and p38 activation, and that TPL-2 inhibition reduces V-ATPase in the phagosomes and the phosphorylation of its regulator, DMXL1. Overall, the manuscript is well written. The finding that TPL-2 controls phagosome acidification and proteolysis independently of MAPK activation is novel and may be potentially interesting. Although most experiments in this study were appropriately designed and executed, the authors need to provide more data to substantiate the manuscript. Overall, the study is preliminary.

We are pleased that this reviewer thought our work was novel and that experiments were appropriately designed and executed. Based on her/his helpful and insightful comments, additional new experiments in the revised manuscript have significantly improved the study as detailed below:

- An important new experiment requested, in which *Dmx1* was knocked down in *Tpl2*[D270A] macrophages (Figure 4E), now shows that the regulation of phagosome maturation by TPL-2 is mediated entirely via DMXL1. Furthermore, expression of 3xFLAG-DMXL1 WT fragment does not rescue the phagosome acidification defect in DMXL1-deficient *Tpl2*[D270A] cells (Figure 4G), in contrast its positive effect in DMXL1-deficient WT cells (Figure 4F). These data are consistent with the 3xFLAG-DMXL1 WT fragment being regulated by TPL-2 catalytic activity.
- V-ATPase function in yeast is controlled by the reversible association of the integral membrane V0 subcomplex with peripheral membrane V1 subcomplex (PMID:22044153). V0/V1 assembly is controlled by the RAVE (regulator of H⁺-ATPase of vacuolar and endosomal membranes) complex, comprised of Skp1p, Rav1p and Rav2p. The mammalian homolog of Rav1, DMXL2 (rabconnectin-3), regulates V-ATPase function in osteoclasts (PMID:20810660), suggesting a conserved function for the RAVE complex in V-ATPase assembly.

TPL-2 regulation of phagosome acidification via DMXL1, a paralog of DMXL2 (54% amino acid identity), suggests that TPL-2 may promote V-ATPase assembly to induce phagosome acidification. Consistent with this possibility, we now show that *Tpl2*^{D270A} mutation significantly reduces the association of ATP6V0D1 (a component of V-ATPase subcomplex V0) with ATP6V1C1 (a component of V-ATPase subcomplex V1) in macrophages incubated with silica beads (Figure 4H).

Unfortunately, we have been unable to carry out some revision experiments that were planned in response to reviewer #1 comments. The very serious COVID19 pandemic situation in London resulted in the the Francis Crick Institute management allowing only essential (COVID19-related) work to be carried out after the Christmas break. In addition, the Ley laboratory closes at the Crick in March, 2021. Nevertheless, we believe that our experiments clearly identify a novel MAP kinase-independent function for TPL-2 regulating phagosome maturation. This provides new insights into the role of TPL-2 signalling in innate immune responses, which likely works in concert with the previously established TPL-2 control of gene expression in macrophages via MAP kinase activation (PMID:19667062, PMID:23842752, PMID:22733995)

Our study provides biochemical and genetic evidence that TPL-2 promotes phagosome acidification via DMXL1. The reviewer suggests several experiments to determine how DMXL1 regulates V-ATPase function. We believe that such experiments are not necessary for this initial demonstration that TPL-2 promotes phagosome maturation in macrophages. There is only one published study with functional experiments on DMXL1, which only shows its requirement for endocytic vesicle acidification in a rat kidney cell line (PMID:26442671). Understanding precisely how DMXL1 regulates V-ATPase function in macrophages should be the focus of a future study. This will require high affinity antibodies for immunoprecipitation and immunoblotting. Such reagents are currently

not available commercially (our unpublished data) and the DMXL1 antibody used by Merkulova et al. (PMID:26442671) does not work on mouse cells (our unpublished data).

Major concerns

Does TPL-2 regulate phagosome maturation and the killing in other phagocytic cells such as neutrophils or dendritic cells?

Macrophage phagosome maturation is essential for effective innate immune responses to bacteria and our current findings clearly establish a role for TPL-2 in regulating this process. Phagosomes also play important roles in neutrophils and dendritic cells (DCs) but are regulated differently to macrophages.

- In contrast to macrophages, neutrophil phagosomes acquire their antimicrobial functions via fusion with secretory vesicles and granules (PMID:21504950). Furthermore, phagosome acidification is not a requirement for neutrophil anti-microbicidal activity.
- Bacterial phagocytosis by DCs is important for antigen presentation and initiation of adaptive immune responses. Compared to macrophages, DC phagosomes are more alkaline, limiting phagosome proteolysis to produce antigens of appropriate size for MHC presentation (PMID:21910624).

In view of these significant differences, the roles of TPL-2 in regulating phagosome function may be distinct between macrophages and neutrophils/DCs. We believe that this should be the subject of future investigations, since this will require extensive analyses for each cell type.

Is the phagocytosis and the killing of other microorganisms (gram-negative, fungus) also controlled by TPL-2?

To address this point, we carried out experiments with the Gram-negative bacterium *Citrobacter rodentium*. We now show that TPL-2 catalytic activity induces acidification of BMDM phagosomes containing *C. rodentium* (Figure 6 F and G) and is required efficient killing of *C. rodentium* by BMDMs (Figure 6E).

Is DMXL1 directly phosphorylated by TPL-2? If not, which kinase is phosphorylating DMXL1?

Consistent with our genetic and pharmacological experiments (Figure 2) showing that TPL-2 regulates DMXL1 phosphorylation independently of MAP kinase activation, the phosphorylation sequence at Ser1903/Ser1904 (AKPCCRGSSFLTSKD**SS**LKLDVREDKCCAAD) does not share similarities with the proline-directed kinase motif targeted by MAP kinases.

To determine initially whether TPL-2 could directly phosphorylate DMXL1 Ser1903/Ser1904, recombinant protein fragments of DMXL1 (containing the target phosphorylation sites) were ordered from the MRC-PPU (Dundee University). We planned to determine whether recombinant TPL-2³⁰⁻⁴⁰⁴ protein (PMID:29229763) was able to phosphorylate DMXL1 Ser1903 *in vitro*. Unfortunately, we were not able to carry out this experiment due to the pandemic lockdown (see above).

Although we were unable to test whether TPL-2 can directly phosphorylate DMXL1, we do not think this experiment, which may have generated negative data, is essential for the paper. Our biochemical and genetic data show clearly that TPL-2 promotes V-ATPase-mediated acidification of phagosomes via DMXL1. The description of this MAP kinase-independent role of TPL-2 is very novel and significantly increases our understanding of the biological processes that TPL-2 regulates in innate immune responses.

Is DMXL1 phosphorylation on S1903 and S1904 modulating V-ATPase recruitment to the phagosome, or V-ATPase activity, or both?

Genetic studies in yeast have demonstrated that reversible disassembly of V0 and V1 subcomplexes is a major mechanism for controlling V-ATPase activity (PMID:22044153). This is regulated by a conserved complex of proteins, which in mammalian cells is called the RAVE complex and includes DMXL2 (rabconnectin-3). DMXL2, which interacts with the cytosolic V1 subcomplex of the V-ATPase, promotes V1 subcomplex association with the membrane-anchored V0 subcomplex to form a catalytically active V0/V1 V-ATPase complex (PMID:20810660).

High similarity with DMXL2 (54% identical) and interaction with the V-ATPase complex suggest that DMXL1 regulates vesicle acidification by controlling V0/V1 assembly (PMID:26442671). We now show that *Tpl2*[D270A] mutation blocks association between ATP6V0D1 (V-ATPase subcomplex 0) and ATP6V1C1 (V-ATPase subcomplex 1) in macrophages incubated with silica beads (new Figure 4H). These data support a model in which TPL-2 catalytic activity induces phagosome acidification by promoting V-ATPase V0/V1 subcomplex assembly via DMXL1 phosphorylation.

How does TPL-2 regulate interactions between V-ATPase and DMXL1? Is DMXL1 phosphorylation on S1903 and S1904 modulating its interaction with V-ATPase?

DMXL1 has been shown to regulate endocytic vesicle acidification (PMID:26442671) but has not been extensively studied and commercially available DMXL1 antibodies do not work on the mouse protein (unpublished data). Consequently, it has not been possible to determine whether TPL-2 regulates DMXL1 association with V-ATPases in macrophages. The function of DMXL1 and how it is regulated by TPL-2 will be a major focus of research in my laboratory in the future.

How are acidification and proteolysis affected in TPL2D270A cells after knocking down DMXL1? And after expressing the 3xFLAG-DMXL1WT? Does 3xFLAG-DMXL1WT also rescue in TPL2D270A cells?

We would like to thank the reviewer for these very good suggestions, which have been addressed with new experiments. Similar to primary BMDMs, *Tpl2*[D270A] mutation reduced phagosome acidification in iBMDMs following uptake of silica beads (Figure EV2B). However, the extent of phagosome acidification was not altered by DMXL1 knockdown (Figure 4E) suggesting that TPL-2 catalytic activity promotes phagosome acidification via DMXL1. Expression of the 3xFLAG-DMXL1 WT fragment in *Tpl2*[D270A] iBMDMs does not rescue phagosome maturation (Figure 4G), in contrast to its effect in WT iBMDMs (Figure 4F). Together these data suggest that TPL-2-mediated phosphorylation is required for DMXL1 to promote phagosome maturation.

What is the effect of phospho-mimicking peptide 3xFLAG-DMXL1SSEE or 3xFLAG-DMXL1SSDD in both TPL2D270A and WT cells?

It is potentially interesting to test the function of SS>EE and SS>DD DMXL1 peptides in regulating phagosome maturation, although such experiments may not be very informative (see below). Unfortunately, due to the Government lockdown and exclusion from the Crick, we were not able to do this.

Replacement of a phospho-site with a negatively charged amino acid often generates negative data, which do not inform about the functional role of a specific phosphorylation. Rather this may result from the inserted negatively charged residue failing to mimic sufficiently a phosphorylated Ser/Thr residue. For example, my laboratory found that S400E mutation of the critical 14-3-3 binding site on the C-terminus of TPL-2 inactivates TPL-2 MEK kinase activity, similar to S400A mutation that blocks 14-3-3 binding (PMID:17709378). However, replacement of Ser400 with R18 peptide to facilitate 14-3-3 binding (in a phosphorylation-independent fashion) reconstitutes TPL-2 MEK kinase activity (PMID:24912162).

Are the DMXL1 peptides (WT and mutants) interacting with V-ATPase? If so, how does this interaction regulate the ATPase activity?

Due to lockdown time constraints, we were unable to determine whether WT/mutant DMXL1 peptides interacted with V-ATPase.

*How are the acidification and the killing of *S. aureus* in WT cells expressing 3xFLAG-DMXL1WT or 3xFLAG-DMXL1SSAA or 3xFLAG-DMXL1SSDD affected? and in the TPL2D270A cells?*

Latex beads have been widely used to investigate the molecular biology of phagosomes (PMID:12892792) and have the advantage that they are inert. In contrast, bacteria have evolved a multitude of strategies to counteract host defences, including impairment of the phagocytic machinery of macrophages (PMID:19369951). Another complicating factor is TLR activation of MAP kinase pathways by bacteria. TPL-2-dependent activation of ERK1/2 and p38 α may alter phagosome maturation indirectly (eg. by changing gene expression) following bacterial infection. For these reasons, we have used beads to establish the mechanism by which TPL-2 catalytic activity promotes phagosome acidification, which resulted in demonstration that TPL-2 promotes phagosome acidification via DMXL1-mediated control of V-ATPase assembly.

Regardless of these these confounding factors, we show that TPL-2 catalytic activity promotes phagosome acidification and cathepsin activation following *Staphylococcus aureus*, similar to our findings with beads. We have not carried out experiments to examine whether TPL-2 controls phagosome acidification via DMXL1 following *S. aureus* infection. However, we believe this is very likely to be the case since the machinery involved in phagosome acidification following uptake of beads is the same as that following uptake of bacteria (PMID:30481638).

Experimental conditions in Figure 1B are not clear. It is the leupeptin included all the time in the assay, and in all conditions?

Figure 1B legend text has been modified to clarify the experimental conditions used. BMDMs were separately pre-treated with 100 μ g/ml leupeptin for 1h to inhibit serine-cysteine proteases. Leupeptin treatment was included as a positive control for the DQ Green BSA/Alexa Fluor 594 bead proteolysis assay.

Many of the experiments in this work are performed using fluorescence beads. How do the authors know that they have cleaned all the external beads attached to the cell membrane?

Phagocytosis uptake assay: cells were washed thoroughly following incubation with beads for 3 min. Cells were then incubated with trypan blue to quench any extracellular fluorescence. This last step ensures that any residual externally attached beads do not fluoresce and fluorescence measured at 30 min reflects only internalised beads.

Phagosome maturation assays: The Trost laboratory has extensive experience with bead-based assays to monitor phagosome maturation (eg. PMID:25504905; PMID:25755298). These well established methods monitor intraphagosomal pH and proteolytic activity (PMID:19590530).

Proteolysis assay: Cells were incubated with DQ Green BSA/Alexa Fluor 594 beads for 3 min and then extensively washed to remove beads that had not been internalised. Fluorescence of DQ green BSA is quenched due to the very high labelling of BSA with many dye molecules. Only after proteolytic cleavage of BSA, the resulting peptides with single dye molecules become fluorescent. Consequently, proteolytic activity associated with any residual external beads is not detected.

pH assay: Cells were incubated with BCECF-beads for 3 min and then extensively washed to remove beads that had not been internalised. The pH-sensitive fluorescent dye BCECF coupled to beads detects the decrease in intraphagosomal pH during maturation. The external cellular milieu is maintained at neutral pH using a buffering solution and consequently fluorescence of non-internalised beads does not change during the course of the assay.

The same question when the authors perform experiments using YFP-S. aureus. This is relevant for the interpretation of the phagocytic uptake results and relevant controls should be shown.

We showed in Figure 5A by CFU measurements that the phagocytic rate, is not dependent on the genotype or bafilomycin treatment. As we used the 'gentamicin protection assay', a standard microbiological method, our findings were not impacted by adherent bacteria. Following uptake of *S. aureus* by BMDMs, the cells were washed several times and then incubated for 1h in culture medium containing 50 µg/ml gentamicin. This kills adhered bacteria that are not internalised, whereas bacteria that are phagocytosed are protected as gentamicin cannot penetrate eukaryotic cells. Continued culture at 10 µg/ml prevents growth of any residual extracellular bacteria without affecting internalised bacteria. CFU measurements at 1 h time point therefore indicated the number of bacteria internalised.

The way the authors represent proteolysis and acidification data is not clear, especially in Fig 2. The colours chosen to represent different experimental conditions are similar and this make very difficult to distinguish from each other. For example: in Fig. 2A and 2B, WT and Nfkb1SSAA data are not clear. In Fig 2D, the WT data cannot be seen. In Fig 2E, the Nfkb1SSAA data cannot be seen.

For increased clarity, the colours for the different conditions have been changed as requested.

In Fig 3, a control showing protein that do not change is missing. Is the decrease in proteins in TPL2D270A samples real? Do TPL2[D270A] cells have less phagosomes?

We have now provided controls that indicating *Tpl2*[D270A] mutation does not induce a total reduction in phagosomes in BMDMs. In Supplementary Figure 1, MS analysis revealed that the abundance of a selection of 16 proteins associated with phagosomes was not significantly different between WT and *Tpl2*^{D270A/D270A} BMDMs. Consistent with this, immunoblotting demonstrated an equivalent amounts of vimentin, an intermediate filament protein that interacts with RAB7 ([PMID:23458836](https://pubmed.ncbi.nlm.nih.gov/23458836/)), in phagosome preparations from WT and *Tpl2*^{D270A/D270A} BMDMs (Figure 3D).

If the killing of S.aureus is diminished in TPL2[D270A] why the cytotoxicity is the same than in WT?

In Figure 6A, CFU is measuring the total number of bacteria in a population of cells and comparing this between WT and *Tpl2*[D270A] cells. CFU numbers, expressed as a fold change from 1 h, will be determined by killing of bacteria by macrophages and growth of bacteria within cells. In addition, if macrophages die by pyroptosis and become permeable following infection, released bacteria will be killed by extracellular antibiotic in the medium.

As expected, we found that *Tpl2*[D270A] mutation decreases the overall killing of *S. aureus*, similar to bafilomycin A treatment. Given that bafilomycin A1 blocks phagosome maturation, our observations imply that an additional killing mechanism is involved in *S. aureus* clearance. In Figure 6C, we show that the amount of host cell death, measured by release of lactate dehydrogenase (LDH) into culture supernatant, is not affected by *Tpl2*[D270A] mutation and is not sufficient to explain the difference in CFU seen in (A). Together these results support the hypothesis that reduced killing by *Tpl2*^{D270A/D270A} BMDMs compared to WT cells was not due to altered host cell death but due to decreased ability of macrophages to kill intracellular bacteria.

It is not clear what does the "n" represent in each figure (field / cells / phagosomes?). The statistical comparison of data in all figures are not clear either.

We have altered the Figure Legends to state explicitly what 'n' represents and clarified the statistical comparisons made.

Minor concerns

Are ERK1/2 and p38 pathways activated after macrophage uptake of latex beads?

Latex beads did not activate ERK1/2 and p38α MAP kinases (data not shown). However, this did not rule out the contribution of very low levels of TPL-2-dependent MAP kinase activation to phagosome

maturation following uptake of latex beads. To investigate this, TPL-2-dependent MAP kinase activation was blocked genetically by *Nfkb1*^{SSAA} mutation and MAP kinase activation blocked pharmacologically. Both of these approaches indicated that TPL-2 activation of MAP kinases was not required to promote phagosome maturation.

The text has been modified to explain the rationale for investigating the potential role of MAP kinase activation in TPL-2 induction of phagosome maturation.

In Fig. 2, the authors should provide blots of total ERK1/2 or p38.

Blots of total ERK1/2 and p38 α have been added to Figure 2 blots as requested.

If TPL-2 controls p38 phosphorylation, why this is not reduced in TPL2KD/Nfkb1SSAA BMDM stimulated with LPS?

My laboratory has previously shown that TPL-2 induces MKK3 and MKK6 activation following LPS stimulation of macrophages, while MKK4 activation is TPL-2 independent (PMID:27402796). Owing to redundancy of MKK3/6 with MKK4, blocking TPL-2 signalling by *Tpl2*[D270A] or *Nfkb1*[SSAA] mutation only fractionally decreases LPS activation of p38 α (between 15-30%).

On page 6 the authors mention: "Nfkb1SSAA mutation, which changes both serines to alanine, blocks IKK-induced p105 proteolysis (Figure 2)". p105 proteolysis is not shown in Fig. 2.

Previous work by my laboratory demonstrated that signal-induced proteolysis of NF- κ B1 p105 is triggered by IKK phosphorylation of p105 Ser930/Ser935 (PMID:12482991) and that this is required for TPL-2 activation of MAP kinase pathways in macrophages (PMID:22733995; PMID:27402796). The aim of the experiment in Figure 2 was to prevent signal-induced TPL-2 activation of MAP kinases by *Nfkb1*[SSAA] mutation, which inhibits IKK-induced p105 proteolysis and prevents TPL-2 phosphorylation of MAP 2-kinases.

As discussed above, latex beads did not detectably activate ERK1/2 and p38 α MAP kinases. Nevertheless, demonstration that *Nfkb1*[SSAA] mutation did not alter phagosome acidification/proteolysis rules out the possibility that low levels of TPL-2-dependent MAP kinase activation promote phagosome maturation.

On page 10: "Consistent with this, S. aureus infection decreased the viability of BMDMs (Figure 6B)" it should be (Figure 6C).

The text has been corrected as suggested.

What does US, UT and NT mean in Fig 4C and F

The text has been modified to define these abbreviations.

Other suggestions

It would be interesting to see if incubation with Bafilomycin of TPL2D270A cells completely revert the effect of TPL2 mutation (Fig 6A).

We thank the reviewer for suggesting this interesting experiment. Unfortunately, due to lockdown time constraints, it was not possible to carry this out.

REVIEWER #2

The study by Breyer and colleagues contains an impressive combination of a variety of techniques to demonstrate that phagosome acidification is strongly dependent on the ability of the TPL-2 kinase to phosphorylate proteins, most notable DMXL1, which is known to associate with the V-ATPase complex. The study has been performed with considerable rigor and the data are compelling and for the most part support the major conclusions made by the authors.

We thank Reviewer 2 for these very positive comments and are pleased that she/he found our data compelling.

Comments

The authors show the reduced acidification through the signal reduction upon protonation of carboxyfluorescein that is coupled to reporter particles. These data are very clear and convincing. However, for ease of analysis and to allow comparison between this study and published studies it would be helpful if the authors converted these into real pH values. This can be achieved using pH standards as detailed in the reference Yates and Russell, 2008, which is already included in the manuscript.

We assayed phagosomal acidification following uptake of BCECF-beads using a reader that monitors single fluorophores. We attempted to switch to a different reader that measures two fluorophores to allow conversion into real pH values. However, we were unable to optimise the assay on this new reader due to problems with low signal sensitivity. Due to lockdown time constraints, we were not able to resolve this technical issue. Nevertheless, we believe our BCECF reporter data clearly demonstrate that TPL-2 catalytic activity is required for phagosome acidification in macrophages (Figure 1B). Furthermore, this conclusion is supported by our imaging data with lysotracker red (Figure 1E).

It would be of value for the authors to utilize reporter beads for the NADPH oxidase activity to generate more quantitative kinetic data, as documented in VanderVen, Yates and Russell, 2009, Traffic 10: 372.

As suggested, we have used reporter beads to monitor NADPH oxidase activity. Our new data confirm that *Tpl2*[D270A] mutation reduces ROS generation (Figure 1F).

The authors show that the generation of reactive oxygen intermediate by the NADPH oxidase complex is reduced, Figure 1F. The link between acidification, phagosome maturation and NADPH oxidase activity is not fully resolved. It is possible that TPL-2 kinase has a direct impact on the formation of the NADPH oxidase complex, do the authors have any data on this? Does blocking acidification with Bafilomycin A impact NADPH oxidase activity? How does this compare with inhibition with C34?

Using reporter beads, we show that Bafilomycin A inhibits ROS generation in BMDMs (Figure 1F), suggesting that the inhibitory effects of *Tpl2*[D270A] mutation of ROS generation are secondary to inhibitory effects on V-ATPase function. RNA sequencing also demonstrated that *Tpl2*[D270A] mutation did not alter the mRNA abundance of *Nox1*, *Noxo1* and *Cyba* components of NADPH oxidase following incubation of BMDMs with LPS-beads (Reviewer #2 response: Figure 1) via inhibition of TPL-2-dependent MAP kinase activation.

LPS-coated beads

Figure 1. TPL-2 catalytic activity does not alter the expression of NADPH oxidase components. Normalised counts for the indicated genes from RNA sequencing of WT and *Tpl2*[D270A] BMDMs following incubation with LPS-coated latex beads for the times shown. Three biological replicates per genotype (n = 3). Data were analysed by a one-way ANOVA for statistically significant differences.

Dear Steve,

Thank you for submitting your revised manuscript to The EMBO Journal. Your study has now been seen by referee #1. As you can see below the referee appreciates the introduced changes and support publication here.

I am therefore very pleased to let you know that we will accept your manuscript for publication here. Before sending you the formal acceptance letter I would like to ask you to sort out the last few comments below:

We can only have 5 keywords - there are currently 7.

Please make sure to make the proteomics data set open. In the data availability section remove username and password

We don't allow data not shown (pg 6). Please rephrase or show the data

The funding information should be included in the Acknowledgement paragraph

The callout to Table 2 (Dataset EV2) is missing.

The legends for tables 1 & 2 should be Dataset EV1 & EV2. Please add the legends as a separate tab in the excel file

We encourage the publication of source data, particularly for electrophoretic gels and blots, with the aim of making primary data more accessible and transparent to the reader. It would be great if you could provide me with a PDF file per figure that contains the original, uncropped and unprocessed scans of all or key gels used in the figure? The PDF files should be labelled with the appropriate figure/panel number, and should have molecular weight markers; further annotation could be useful but is not essential. The PDF files will be published online with the article as supplementary "Source Data" files.

We include a synopsis of the paper that is visible on the html file (see <http://emboj.embopress.org/>). Could you provide me with a general summary statement and 3-5 bullet points that capture the key findings of the paper?

It would also be good if you could provide me with a summary figure that I can place in the synopsis. The size should be 550 wide by 400 high (pixels).

I have asked our publisher to do their prepublication checks on the paper. They will send me the file within the next few days. Please wait to upload the revised version until you have received their comments.

When you submit the revised version please also upload a point-by-point response

That should be all - congratulations on a nice study!

with best wishes

Karin

Karin Dumstrei, PhD
Senior Editor
The EMBO Journal

Further information is available in our Guide For Authors:

The revision must be submitted online within 90 days; please click on the link below to submit the revision online before 3rd Jun 2021.

Referee #1:

In the revised version, the authors have implemented many of the suggested reviewer comments,

made some valuable additions to an overall good manuscript and further improved on the clarity of the manuscript. The point by point response is very well argued.
Therefore, I now recommend publication of this study in the EMBO Journal.

Dear Steve,

Thank you for submitting your revised manuscript to The EMBO Journal. I have now had a chance to take a careful look at everything and all looks good.

I am therefore very pleased to accept the manuscript for publication here.

Congratulations on a great study!

with best wishes

Karin

Karin Dumstrei, PhD
Senior Editor
The EMBO Journal

Please note that it is EMBO Journal policy for the transcript of the editorial process (containing referee reports and your response letter) to be published as an online supplement to each paper. If you do NOT want this, you will need to inform the Editorial Office via email immediately. More information is available here: https://emboj.embopress.org/about#Transparent_Process

Your manuscript will be processed for publication in the journal by EMBO Press. Manuscripts in the PDF and electronic editions of The EMBO Journal will be copy edited, and you will be provided with page proofs prior to publication. Please note that supplementary information is not included in the proofs.

Should you be planning a Press Release on your article, please get in contact with embojournal@wiley.com as early as possible, in order to coordinate publication and release dates.

If you have any questions, please do not hesitate to call or email the Editorial Office. Thank you for your contribution to The EMBO Journal.

Corresponding Author Name: Prof. Steven Ley

Journal Submitted to: The EMBO Journal

Manuscript Number: EMBOJ-2020-106188